



**A bulk-mass-modeling-based method for retrieving Particulate Matter Pollution using**
**CALIOP observations**

Travis D. Toth[1], Jianglong Zhang[2], Jeffrey S. Reid[3], and Mark A. Vaughan[1]
[1]NASA Langley Research Center, Hampton, VA
[2]Department of Atmospheric Sciences, University of North Dakota, Grand Forks, ND
[3]Marine Meteorology Division, Naval Research Laboratory, Monterey, CA
*Correspondence to*: Travis D. Toth (travis.d.toth@nasa.gov); Jianglong Zhang
(jianglong.zhang@und.edu)
**Abstract.**  In this proof-of-concept paper, we apply a bulk-mass-modeling method using
observations from the NASA Cloud-Aerosol Lidar with Orthogonal Polarization (CALIOP)
instrument for retrieving particulate matter (PM) concentration over the contiguous United States
(CONUS) over a 2-year period (2008-2009).  Different from previous approaches that rely on
empirical relationships between aerosol optical depth (AOD) and $PM_{2.5}$ (PM with particle sizes
less than 2.5 μm), for the first time, we derive $PM_{2.5}$ concentrations, both at daytime and nighttime,
from near surface CALIOP aerosol extinction retrievals using bulk mass extinction coefficients
and model-based hygroscopicity.  Preliminary results from this 2-year study conducted over the
CONUS show a good agreement ($r^2 \sim 0.48$; mean bias of -3.3 μg m$^{-3}$) between the averaged
nighttime CALIOP-derived $PM_{2.5}$ and ground-based $PM_{2.5}$ (with a lower $r^2$ of ~0.21 for daytime;
mean bias of -0.4 μg m$^{-3}$), suggesting that PM concentrations can be obtained from active-based
spaceborne observations with reasonable accuracy.  Results from sensitivity studies suggest that
accurate aerosol typing is needed for applying CALIOP measurements for $PM_{2.5}$ studies.  Lastly,
the e-folding correlation length for surface $PM_{2.5}$ is found to be around 600 km for the entire
CONUS (~300 km for Western CONUS and ~700 km for Eastern CONUS), indicating that




CALIOP observations, although sparse in spatial coverage, may still be applicable for PM$_{2.5}$
studies.
**1      Introduction**
During the last decade, an extensive number of studies have researched the feasibility of
estimating PM$_{2.5}$ (particulate matter with particle sizes smaller than 2.5 μm) pollution with the use
of passive-based satellite-derived aerosol optical depth (AOD; e.g., Liu et al., 2007; Hoff and
Christopher, 2009; van Donkelaar et al., 2015). Monitoring of PM concentration from space
observations is needed, as PM$_{2.5}$ pollution is one of the known causes of respiratory related diseases
as well as other health related issues (e.g., Liu et al., 2005; Hoff and Christopher, 2009; Silva et
al., 2013). Yet, ground-based PM$_{2.5}$ measurements are often inconsistent or have limited
availability over much of the globe.
In some earlier studies, empirical relationships of PM$_{2.5}$ concentrations and AODs were
developed and used for estimating PM$_{2.5}$ concentrations from passive sensor retrieved AODs (e.g.,
Wang and Christopher, 2003; Engel-Cox et al., 2004; Liu et al., 2005; Kumar et al., 2007; Hoff
and Christopher, 2009). One of the limitations of this approach is that vertical distributions and
thermodynamic state of aerosol particles vary with space and time. Especially for regions with
elevated aerosol plumes, deep boundary layer entrainment zones, or strong nighttime inversions,
column-integrated AODs are not a good approximation of surface PM$_{2.5}$ concentrations at specific
points and times (e.g., Liu et al., 2004; Toth et al., 2014; Reid et al., 2017). Indeed, Kaku et al.
(2018) recently showed that surface PM$_{2.5}$ had longer spatial correlation lengths than AOD, even
in the "well behaved" southeastern United States where previous studies showed good
performance (e.g., Wang and Christopher, 2003). To account for variability in aerosol vertical




distribution, several studies have attempted the use of chemical transport models, or CTMs (e.g.,
van Donkelaar et al., 2015).  Satellite data assimilation of AOD has become commonplace, vastly
improving AOD analyses and short-term prediction (e.g., Zhang et al., 2014; Sessions et al., 2015).
Yet, PM$_{2.5}$ simulations remain poor (e.g., Reid et al., 2016).  Uncertainties in such studies are
unavoidable due to uncertainties in CTM-based aerosol vertical distributions, and no nighttime
AODs are currently available from passive-based satellite retrievals.

It is arguable that from a climatological/long-term average perspective, the use of AOD as

a proxy for PM$_{2.5}$ concentrations nevertheless has certain qualitative skill (e.g., Toth et al., 2014;
Reid et al., 2017) for the most significant events as well as due to the averaging process that
suppresses sporadic aerosol events with highly variable vertical distributions.  Still, as illustrated
in Fig. 1, where 2-year (2008-2009) means of Moderate Resolution Imaging Spectroradiometer
(MODIS) AOD are plotted against PM$_{2.5}$ concentrations throughout the contiguous United States
(CONUS), although a linear relationship is plausibly shown, a low r$^2$ value of 0.09 is found.  To
construct Fig. 1, Aqua MODIS Collection 6 (C6) Optical_Depth_Land_And_Ocean data (0.55
μm), restricted to "Very Good" retrievals as reported by the Land_Ocean_Quality_Flag, are first
collocated with daily surface PM$_{2.5}$ measurements in both space and time (i.e., within 40 km in
distance and the same day), and then collocated daily pairs are averaged into 2-year means (for
each PM$_{2.5}$ site).  Figure 1 may be indicating that even from a long-term mean perspective, aerosol
vertical distributions are not uniform across the CONUS, which is also confirmed by other studies
(e.g., Toth et al., 2014).  AOD retrievals themselves, with known uncertainties due to cloud
contamination and assumptions in the retrieval process (e.g., Levy et al., 2013), may also introduce
uncertainties to that task.





On board the Cloud-Aerosol Lidar and Infrared Pathfinder Satellite Observations
(CALIPSO) satellite, the Cloud-Aerosol Lidar with Orthogonal Polarization (CALIOP) instrument
provides observations of aerosol and cloud vertical distributions at both day and night (Hunt et al.,
2009; Winker et al., 2010).  Given that CALIOP provides aerosol extinction retrievals near the
ground, it is interesting and reasonable to raise the question: can near surface CALIPSO extinction
be used as a better physical quantity than AOD for estimating surface $PM_{2.5}$ concentrations?  This
is because unlike AOD, which is a column-integrated value, near surface CALIPSO extinction is,
in theory, a more realistic representation of near surface aerosol properties.  Yet, in comparing
with passive sensors such as MODIS, which has a swath width on the order of ~2000 km, CALIOP
is a nadir pointing instrument with a narrow swath of ~70 m and a repeat cycle of 16 days (Winker
et al., 2009).  Thus, the spatial sampling of CALIOP is sparse on a daily basis and temporal
sampling or other conditional or contextual biases are unavoidable if CALIOP observations are
used to estimate daily $PM_{2.5}$ concentrations (Zhang and Reid, 2009; Colarco et al., 2014).  Also,
there are known uncertainties in CALIPSO retrieved extinction values due to uncertainties in the
retrieval process, such as the lidar ratio (extinction-to-backscatter ratio), calibration, and the
"retrieval fill value" (RFV) issue (Young et al., 2013; Toth et al., 2018).
Even with these known issues, especially the sampling bias, it is still compelling to
investigate if near surface CALIOP extinction can be utilized to retrieve surface $PM_{2.5}$
concentrations with reasonable accuracy from a long-term (i.e., two-year) mean perspective.
CALIOP data have been successfully used in $PM_{2.5}$ studies in the past, but primarily for assisting
passive-based AOD/$PM_{2.5}$ analyses using aerosol vertical distribution as a constraint (e.g., Glantz
et al., 2009; van Donkelaar et al., 2010; Val Martin et al., 2013; Toth et al., 2014; Li et al., 2015;
Gong et al., 2017).  However, the question remained as to the efficacy of the direct use of CALIOP



retrievals. To demonstrate a concept, we developed a bulk mass scattering scheme for inferring
PM concentrations from near surface aerosol extinction retrievals derived from CALIOP
observations. The bulk method used here is based upon the well-established relationship between
particle light scattering and $PM_{2.5}$ aerosol mass concentration (e.g., Charlson et al., 1968;
Waggoner and Weiss, 1980; Liou, 2002; Chow et al., 2006), discussed further, with the relevant
equations, in Sect. 2.
In this study, using two years (2008-2009) of CALIOP and United States (U.S.)
Environmental Protection Agency (EPA) data over the CONUS, the following questions are
addressed:
1. Can CALIOP extinction be used effectively for estimating $PM_{2.5}$ concentrations through a
bulk mass scattering scheme from a 2-year mean perspective for both daytime and
nighttime?
2. Can CALIOP extinction be used as a better parameter than AOD for estimating $PM_{2.5}$
concentrations from a 2-year mean perspective?
3. What are the sampling biases we can expect in CALIOP estimates of $PM_{2.5}$?
4. How do uncertainties in bulk properties compare to overall CALIOP-retrieved $PM_{2.5}$
uncertainty?
Details of the methods and datasets used are described in Sect. 2. Section 3 shows the
preliminary results using two years of EPA $PM_{2.5}$ and CALIOP data, including an uncertainty
analysis. The conclusions of this paper are provided in Sect. 4.





## 2 Data and Methods


Since 1970, the U.S. EPA has monitored surface PM using a number of Federal
Reference/Equivalent Methods (FRMs/FEMs), which employ gravimetric, tapered element
oscillating microbalance (TEOM), and beta gauge instruments (Federal Register, 1997;
Greenstone, 2002). Two years (2008-2009) of daily $PM_{2.5}$ Local Conditions (EPA code = 88101)
data were acquired from the EPA Air Quality System for use in this investigation, consistent with
our previous $PM_{2.5}$ study (Toth et al., 2014). We note that these data represent $PM_{2.5}$ concentrations
over a 24-hour period and include two scenarios: one sample is taken during the 24-hour duration
(i.e., filter-based measurement), or an average is computed from hourly samples within this time
period (every hour may not have an available measurement, however).
CALIOP, flying aboard the CALIPSO platform within the A-Train satellite constellation,
is a dual wavelength (0.532 and 1.064 μm) lidar that has collected profiles of atmospheric aerosol
particles and clouds since summer 2006 (Winker et al., 2007). In this study, daytime and nighttime
extinction coefficients retrieved at 0.532 μm from the Version 4.10 CALIOP Level 2 5 km aerosol
profile (L2_05kmAPro) product were used. Using parameters provided in the L2_05kmAPro
product, as well as the corresponding Level 2 5 km aerosol layer (L2_05kmALay) product, a robust
quality-assurance (QA) procedure for the aerosol observations was implemented (Table 1).
Further information on the QA metrics and screening protocol are discussed in detail in previous
studies (Kittaka et al. 2011; Campbell et al. 2012; Toth et al. 2013; 2016). Once the QA procedure
was applied, the aerosol profiles were linearly re-gridded from 60 m vertical resolution (above
mean sea level [AMSL]) to 100 m segments (i.e., resampled to 100 m resolution) referenced to the
local surface (above ground level [AGL]; Toth et al., 2014; 2016). The choice of 100 m was
arbitrary, and the profiles were re-gridded in order to obtain an AGL-corrected dataset, as opposed



to the AMSL-referenced profiles provided by the L2_05kmAPro product.  Surface elevation and
relative humidity (RH) were taken from collocated model data included in the CALIPSO
L2_05kmAPro product (CALIPSO Data Products Catalog (Release 4.20); RH taken from the
Modern Era Retrospective-Analysis for Research, or MERRA-2 reanalysis product).  To limit the
effects of signal attenuation and increase the chances of measuring aerosol presence near the
surface, the Atmospheric Volume Description parameter within the L2_05kmAPro dataset is used
to cloud-screen each aerosol profile as in Toth et al. (2018).

In this study, near surface PM mass concentration ($C_m$) is derived from near surface

CALIOP extinction based on a bulk formulation as in Equation 1 (e.g., Liou, 2002; Chow et al.,

2006):

$$\beta = C_m(a_{scat}f_{rh} + a_{abs}) \text{ x } 1000 \qquad (1)$$

where $\beta$ is CALIOP-derived near surface extinction in km$^{-1}$, $C_m$ is the PM mass

concentration in µg m$^{-3}$, $a_{scat}$ and $a_{abs}$ are dry mass scattering and absorption efficiencies in m$^2$ g$^{-1}$,
and $f_{rh}$ represents the light scattering hygroscopicity, respectively.  As a preliminary study, for the
purpose of demonstrating this concept, we assume the dominant aerosol type over the contiguous
U.S. (CONUS) is pollution aerosol (i.e., the most prevalent near-surface aerosol type reported in
the CALIOP products for the CONUS during 2008-2009 is polluted continental) with $a_{scat}$ and $a_{abs}$
values of 3.40 and 0.37 m$^2$ g$^{-1}$ (Hess et al., 1998; Lynch et al., 2016), respectively.  These values
are similar to those reported in Malm and Hand (2007) and Kaku et al. (2018) but are interpolated
to 0.532 µm from values at 0.450 µm and 0.550 µm obtained from the Optical Properties of
Aerosols and Clouds (OPAC) model (Hess et al., 1998).  Still, both $a_{scat}$ and $a_{abs}$ have regional and
species related dependencies.  Also, only 2-year averages are used in this study, and we assume
that sporadic aerosol plumes are smoothed out in the averaging process, and that bulk aerosol



properties are similar throughout the study region. We have further explored the impact of aerosol
types to $PM_{2.5}$ retrievals in a later section. Furthermore, to aid in focusing this study on fine
mode/anthropogenic aerosols, those aerosol extinction range bins classified as dust by the CALIOP
typing algorithm were excluded from the analysis.
Also, surface PM concentrations are dry mass measurements. To account for the impact
of humidity on $a_{scat}$ (it is assumed that $a_{abs}$ is not affected by moisture), we estimated the
hygroscopic growth factor for pollution aerosol based on Hanel (1976), as shown in Equation 2:

$$f_{rh} = (\frac{1-RH}{1-RH_{ref}})^{-\Gamma}$$

(2)

where $f_{rh}$ is the hygroscopic growth factor, $RH$ is the relative humidity, and $RH_{ref}$ is the
reference RH and is set to 30% in this study (Lynch et al., 2016). $\Gamma$ is a unitless value (a fit
parameter describing the amount of hygroscopic increase in scattering) and is assumed to be 0.63
(i.e., sulfate aerosol) in this study (Hanel, 1976; Chew et al., 2016; Lynch et al., 2016).
Lastly, the CALIOP-derived PM density is for all particle sizes. To convert from mass
concentration of PM ($C_m$) to mass concentration of $PM_{2.5}$ ($C_{m2.5}$), which represents mass
concentration for particle sizes smaller than 2.5 μm, we adopted the $PM_{2.5}$ to $PM_{10}$ (PM with
diameters less than 10 μm) ratio ($\phi$) of 0.6 as measured during the Studies of Emissions and
Atmospheric Composition, Clouds and Climate Coupling by Regional Surveys (SEAC⁴RS)
campaign over the US (Kaku et al., 2018). Again, the ratio of $PM_{2.5}$ to $PM_{10}$ can also vary spatially,
however we used a regional mean to demonstrate the concept. Analyses in a later section using
two-years (2008-2009) of surface $PM_{2.5}$ to $PM_{10}$ data suggest that 0.6 is a rather reasonable number
to use for the CONUS for the study period. Here we assume that mass concentrations for particle
sizes larger than 10 μm are negligible over the CONUS. Thus, we can rewrite Equation 1 as:




$$C_{m2.5} = \frac{\beta \times \phi}{(a_{scat} \times f_{rh} + a_{abs}) \times 1000}$$
(3)

where $C_{m2.5}$ is the CALIOP-derived PM$_{2.5}$ concentration in units of μg m$^{-3}$.

**3**      **Results and Discussion**
**3.1 Regional analysis**
Figure 2a shows the mean PM$_{2.5}$ concentration using two years (2008-2009) of daily
surface PM$_{2.5}$ data from the U.S. EPA (PM$_{2.5\_EPA}$), not collocated with CALIOP observations. A
total of 1,091 stations (some operational throughout the entire period; others only partially) are
included in the analysis and observations from those stations are further used in evaluating
CALOP-derived PM$_{2.5}$ concentrations (C$_{m2.5}$), as later shown in Fig. 3. PM$_{2.5}$ concentrations of
~10 μg m$^{-3}$ are found over the eastern CONUS. In comparison, much lower PM$_{2.5}$ concentrations
of ~5 μg m$^{-3}$ are exhibited for the interior CONUS, over states including Montana, Wyoming,
North Dakota, South Dakota, Utah, Colorado, and Arizona. For the west coast of the CONUS,
and especially over California, higher PM$_{2.5}$ concentrations are observed, with the maximum two-
year mean near 20 μg m$^{-3}$. Note that the spatial distribution of surface PM$_{2.5}$ concentrations over
the CONUS as shown in Fig. 2a is consistent with reported values from several studies (e.g., Hand
et al., 2013; Van Donkelaar et al., 2015; Di et al., 2017).
Figure 3a shows the two-year averaged 1° x 1° (latitude/longitude) gridded daytime
CALIOP aerosol extinction over the CONUS using CALIOP observations from 100-1000 m,
referenced to the number of cloud-free L2_05kmAPro profiles in each 1 x 1° bin. The lowest 100
m of CALIOP extinction data are not used in the analysis due to the potential of surface return
contamination (e.g., Toth et al., 2014), although this has been improved for the Version 4 CALIOP
products but may still be present in some cases. Here the averaged extinction from 100-1000 m is



used to represent near surface aerosol extinction. This selection of the 100-1000 m layer is
somewhat arbitrary, even though it is estimated from the mean CALIOP-based aerosol vertical
distribution over the CONUS (Toth et al., 2014), as surface layer heights may change seasonally
and diurnally. Thus, a sensitivity study is provided in a later section to understand the impact of
this aerosol layer selection to CALIOP-based $PM_{2.5}$ retrievals. As shown in Fig. 3a, higher mean
near surface CALIOP extinction of 0.1 km$^{-1}$ are found for the eastern CONUS and over California,
while lower values of 0.025-0.05 km$^{-1}$ found for the interior CONUS. Figure 3b shows a plot
similar to Fig. 3a but using nighttime CALIOP observations only. Although similar spatial
patterns are found during both day and night, the near surface extinction values are overall lower
for nighttime than daytime, and nighttime data are less noisy than daytime. These findings are not
surprising, as daytime CALIOP measurements are subject to contamination from background solar
radiation (e.g., Omar et al., 2013).

To investigate any diurnal biases in the data, Figs. 3c and 3d show the derived $PM_{2.5}$

concentration using daytime and nighttime CALIOP data respectively, based on the method
described in Section 2. Both Figures 3c and 3d suggest a higher $PM_{2.5}$ concentration of ~10-12.5
μg m$^{-3}$ over the eastern CONUS, and a much lower $PM_{2.5}$ concentration of ~2.5-5 μg m$^{-3}$ over the
interior CONUS. High $PM_{2.5}$ values of 10-20 μg m$^{-3}$ are also found over the west coast of the
CONUS, particularly over California. The spatial distribution of $PM_{2.5}$ concentrations, as derived
using near surface CALIOP data (Figs. 3c and 3d, as well as the combined daytime and nighttime
perspective shown in Fig. 2c), is remarkably similar to the spatial distribution of $PM_{2.5}$ values as
estimated based on ground-based observations (Fig. 2a). Still, day and night differences in $PM_{2.5}$
concentrations are also clearly visible, as higher $PM_{2.5}$ values are found, in general, during daytime,
based on CALIOP observations. The high daytime $PM_{2.5}$ values, as shown in Fig. 3c, may



represent stronger near surface convection and more frequent anthropogenic activities during
daytime. However, they may also be partially contributed from solar radiation contamination.
Another possibility is that the daytime mean extinction coefficients (from which the mean $PM_{2.5}$
estimates are derived) appear artifically larger than at night due to high daytime noise limiting the
ability of CALIOP to detect fainter aerosol layers during daylight operations.
Figure 3e shows the inter-comparison between $PM_{2.5\_EPA}$ and $PM_{2.5\_CALIOP}$ concentrations.
Note that only CALIOP and ground-based $PM_{2.5}$ data pairs, which are within 100 km of each other
and have reported values for the same day (i.e., year, month, and day), are used to generate Fig.
3e. Still, although only spatially and temporally collocated data pairs are used, ground-based $PM_{2.5}$
data represent 24-hour averages, while CALIOP-derived $PM_{2.5}$ concentrations are instantaneous
values over the daytime CALIOP overpass. To reduce this temporal bias, two years (2008-2009)
of collocated CALIOP-derived and measured $PM_{2.5}$ concentrations are averaged and only the two-
year averages are used in constructing Fig 3e. Also, to minimize the above-mentioned temporal
sampling bias, ground stations with fewer than 100 collocated pairs are discarded. This leaves a
total of 276 stations for constructing Fig. 3e.
As shown in Fig. 3e, an $r^2$ value of 0.21 (with a slope of 0.48) is found between CALIOP-
derived and measured surface $PM_{2.5}$ concentrations, with a corresponding mean bias of -0.40 µg
$m^{-3}$ ($PM_{2.5\_CALIOP}$ - $PM_{2.5\_EPA}$). In comparison, Fig. 3f shows results similar to Fig. 3e, but for
nighttime CALIOP data. A much higher $r^2$ value of 0.48 (with a slope of 0.67) is found between
CALIOP-derived and measurement $PM_{2.5}$ values from 528 EPA stations, with a corresponding
mean bias of -3.3 $µgm^{-3}$ ($PM_{2.5\_CALIOP}$ - $PM_{2.5\_EPA}$). This may be related to the diurnal variability
of $PM_{2.5}$ concentrations, as the daily mean EPA measurement might be closer to the CALIOP A.M.
retrieval than to its P.M. counterpart. Still, data points are more scattered in Fig. 3e in comparison



with Fig. 3f, which again indicates that daytime CALIOP data are noisier, possibly due to daytime
solar contamination as well as other factors such as biases in relative humidity. Details of these
biases are further explored in Section 3.2.
To supplement this analysis, a pairwise $PM_{2.5\_EPA}$ and $PM_{2.5\_CALIOP}$ (day and night CALIOP
combined) analysis is presented in the spatial plots of Figs. 2b and 2d. Here, however, we lift the
100 collocated pairs requirement to increase data samples for better spatial representativeness. The
spatial variability of $PM_{2.5}$ over the CONUS is consistent with the observed patterns of non-
collocated data (i.e., Figs. 2a and 2c), but with generally higher values due to differences in
sampling. Also, comparing Figs. 2b and 2d, $PM_{2.5\_EPA}$ spatial patterns match well with those of
$PM_{2.5\_CALIOP}$, yet with larger values for $PM_{2.5\_EPA}$ (consistent with the biases discussed above).
Lastly, a scatterplot of the pairwise analysis shown in Figs. 2b and 2d is provided in Fig. 4. An $r^2$
value of 0.40 is found between EPA and CALIOP-derived $PM_{2.5}$ concentrations from a combined
daytime and nighttime CALIOP perspective. Overall, Figs. 2, 3, and 4 indicate that near surface
CALIOP extinction data can be used to estimate surface $PM_{2.5}$ concentrations with reasonable
accuracy.

**3.2 Uncertainty analysis**
In this section, uncertainties in the CALIOP derived, 2-year averaged $PM_{2.5}$ concentrations
are explored as functions of aerosol vertical distribution, $PM_{2.5}$ to $PM_{10}$ ratio, RH, aerosol type,
and cloud presence above. Spatial sampling related biases as well as prognostic errors are also
studied.






**3.2.1 Prognostic errors in $C_{m2.5}$**
As a first step for the uncertainty analysis, we estimated the prognostic error of 2-year
averaged $PM_{2.5\_CALIOP}$. Figure 5 shows the root-mean-square error (RMSE) of CALIOP-based
$PM_{2.5}$ concentrations against those from EPA stations as a function of CALIOP-based $PM_{2.5}$ for
the 2008-2009 period over the CONUS. RMSEs were computed in intervals of 5 µg m$^{-3}$ from 0
to 25 µg m$^{-3}$, with no computations greater than 25 µg m$^{-3}$ performed due to very few data points
above this $PM_{2.5}$ concentration level. A mean combined daytime and nighttime minimum error of
~4 µg m$^{-3}$ is found, with generally larger RMSEs for nighttime below 15 µg m$^{-3}$, and larger RMSEs
for daytime above 15 µg m$^{-3}$. However, mean RMSEs (i.e., computed from the RMSEs shown in
Fig. 5) are similar for both datasets, ~4.5 µg m$^{-3}$ for daytime and ~4.0 µg m$^{-3}$ for nighttime. Also,
note that while the absolute error for daytime is largest at high $PM_{2.5}$ concentrations, relative errors
are similar (e.g., 3 µg m$^{-3}$/10 µgm$^{-3}$ or 30% for the 5-10 µg m$^{-3}$ bin, versus 7 µg m$^{-3}$/25 µg m$^{-3}$ or
28% for the 20-25 µg m$^{-3}$ bin). For context, the number of samples per bin are also plotted (as X
symbols) in Fig. 5. Data sample sizes are smallest (largest) for the lowest/highest range (mid-
range) $PM_{2.5}$ bins.

**3.2.2 Surface layer height sensitivity study**
A sensitivity study was conducted for which $PM_{2.5}$ was derived from near-surface CALIOP
aerosol extinction by varying the height of the surface layer in increments of 100 m from the
ground to 1000 m. Note that the surface layer (0-100 m) is included for this sensitivity study only.
The statistical results of this analysis, for both daytime and nighttime conditions, are shown in
Table 2. Four statistical parameters were computed, consisting of $r^2$, slope, mean bias (CALIOP
– EPA) of $PM_{2.5}$, and percent error change in derived $PM_{2.5}$, defined as: ((mean_new_$PM_{2.5}$ –



mean_original_PM$_{2.5}$)/mean_original_PM$_{2.5}$)*100.  For context, the bottom row of Table 2 shows
the results from the original analysis.  In terms of r$^2$ and slope, optimal values peak at different
surface layer heights between daytime and nighttime.  For example, for daytime, the largest
correlations are found for the 0-600 m and 0-700 m layers, while for nighttime these are found for
the 0-300 m and 0-400 m layers.  However, the 0-300 m layer (100-1000 m layer) exhibits the
lowest mean bias for the daytime (nighttime) analysis.  Overall, marginal changes are found for
varying the height of the surface layer.  Yet the largest mean bias is found for the 0-100 m layer,
indicating the need for excluding the 0-100 m layer in the analysis.

**3.2.3 RH sensitivity study**
Profiles of RH were taken from the MERRA-2 reanalysis product, as these collocated data
are provided in the CALIPSO L2_05kmAPro product.  However, biases may exist in this RH
dataset.  Thus, we examined the impact of varying the RH values by +/- 10% on the CALIOP-
derived PM$_{2.5}$ concentrations.  For both daytime and nighttime analyses, no significant differences
in the r$^2$ and slope values were found.  However, a +15% (-15%) change in the mean derived PM$_{2.5}$
values was found by decreasing (increasing) the RH values by 10%.

**3.2.4 PM$_{2.5}$ to PM$_{10}$ ratio sensitivity study**
Another source of uncertainty in this study is the PM$_{2.5}$/PM$_{10}$ ratio.  Using surface-based
PM$_{2.5}$ and PM$_{10}$ data from those EPA stations over the CONUS for 2008-2009 with concurrent
PM$_{2.5}$ and PM$_{10}$ daily data available (i.e., 409 stations), we computed the mean PM$_{2.5}$/PM$_{10}$ ratio,
and its corresponding standard deviation.  The mean ratio was 0.56 with a standard deviation of
0.32.  It is interesting to note that the mean PM$_{2.5}$/PM$_{10}$ ratio estimated from two years of surface





observations over the CONUS is close to 0.6 (the number used in this study), as reported by Kaku
et al. (2018).  We also tested the sensitivity of the derived $PM_{2.5}$ concentrations as a function of
$PM_{2.5}/PM_{10}$ ratio for two scenarios: ±1 standard deviation of the mean (Table 3).  In general, a ±50
% to 60 % change is found with the variation of the $PM_{2.5}/PM_{10}$ ratio at the range of ±1 standard
deviation of the mean.  As suggested from Table 3, the optimal slope is found using a ratio of +1
standard deviation of the mean for both daytime and nighttime.  The lowest mean daytime bias is
found for a ratio of 0.6, and for nighttime the lowest mean bias occurs using a ratio of 0.88.

**3.2.5 Sampling-related biases**

As mentioned in the introduction section, a sampling bias, due to the very small footprint

size and ~16 day repeat cycle of CALIOP, can exist when using CALIOP observations for $PM_{2.5}$
estimates (Zhang and Reid, 2009).  This sampling-induced bias is investigated from a 2-year mean
perspective by comparing histograms of $PM_{2.5\_EPA}$ and $C_{m2.5}$ concentrations as shown in Fig. 6.  To
generate Fig. 6, all available daily EPA $PM_{2.5}$ are used to represent the "true" 2-year mean spectrum
of $PM_{2.5}$ concentrations over the EPA sites.  The aerosol extinction data spatially collocated to the
EPA sites (Sect. 3.1), but not temporally collocated, are used for estimating the 2-year mean
spectrum of $PM_{2.5}$ concentrations as derived from CALIOP observations.  To be consistent with
the previous analysis, only cloud-free CALIOP profiles are considered.    The $PM_{2.5\_EPA}$
concentrations peak at ~9 µg m$^{-3}$ (standard deviation of ~3 µg m$^{-3}$), and CALIOP-derived $PM_{2.5}$
peaks at ~9 µg m$^{-3}$ (daytime; standard deviation of ~4 µg m$^{-3}$) and ~5 µg m$^{-3}$ (nighttime; standard
deviation of ~2 µg m$^{-3}$).  The distribution shifts towards smaller concentrations for CALIOP, more
so for nighttime than daytime (possibly due to CALIOP daytime versus nighttime detection
differences).




Still, Fig. 6 may reflect the diurnal difference in $PM_{2.5}$ concentrations as well as the
retrieval bias in $C_{m2.5}$ values.  Thus, we have re-performed the exercise shown in Fig. 6 using
spatially and temporally collocated $PM_{2.5\_EPA}$ and $C_{m2.5}$ data as shown in Fig. 7.  To construct Fig.
7, $PM_{2.5\_EPA}$ and $C_{m2.5}$ data are collocated following the steps mentioned in Sect. 3.1, with CALIOP
and EPA $PM_{2.5}$ representing 2-year mean values for each EPA station.  Again, only cloud-free
CALIOP profiles are considered for this analysis.  As shown in Fig. 7a, the $PM_{2.5\_EPA}$
concentrations peak at ~7 μg m$^{-3}$ (standard deviation of ~4 μg m$^{-3}$), and daytime $C_{m2.5}$ peaks at
~6 μg m$^{-3}$ (standard deviation of ~4 μg m$^{-3}$).  In comparison, with the use of collocated nighttime
$C_{m2.5}$ and $PM_{2.5\_EPA}$ data as shown in Fig. 7b, the peak $PM_{2.5\_EPA}$ value is about 2 μg m$^{-3}$ higher
than the peak $C_{m2.5}$ value (with similar standard deviations as found in the analyses of Fig. 7a).
Considering both Figs. 6 and 7, it is likely that the temporal sampling bias seen in Fig. 6 is at least
in part due to retrieval bias as well as the difference in $PM_{2.5}$ concentrations during daytime and
nighttime.

**3.2.6 CALIOP AOD analysis**
Most past studies focused on the use of column AODs as proxies for surface $PM_{2.5}$ (e.g.,
Liu et al., 2005; Hoff and Christopher, 2009; van Donkelaar et al., 2015).  Therefore, it is
interesting to investigate whether near surface CALIOP extinction values can be used as a better
physical quantity to estimate surface $PM_{2.5}$ in comparing with column-integrated CALIOP AOD.
To achieve this goal, we have compared CALIOP column AOD and $PM_{2.5}$ from EPA stations, as
shown in Fig. 8.  Similar to the scatterplots of Fig. 4, each point represents a two-year mean for
each EPA site, and was created from a dataset following the same spatial/temporal collocation as
described above.  As shown in Fig. 9, $r^2$ values of 0.04 and 0.13 are found using CALIOP daytime





and nighttime AOD data, respectively, similar to the MODIS-based analysis shown in Fig. 1. This
is expected, as elevated aerosol layers will negatively impact the relationship between surface
$PM_{2.5}$ and column AOD. The derivation of surface $PM_{2.5}$ from near surface CALIOP extinction,
as demonstrated from this study however, provides a much better spatial matching between the
quantities being compared, with potential error terms that can be well quantified and minimized in
later studies.

**382  3.2.7 Cloud flag sensitivity study**

For most of this paper, a strict cloud screening process is implemented, during which no

clouds are allowed in the entire CALIOP profile. However, in contrast to passive sensor
capabilities (e.g., MODIS), near-surface aerosol extinction coefficients can be readily retrieved
from CALIOP profiles even when there are transparent cloud layers above. Therefore, we
conducted an additional analysis for which no cloud flag was set (i.e., all-sky conditions). Results
are shown in scatterplot form in Fig. 9, in a similar manner as Figs. 3e and f, with an additional 97
(156) points for the daytime (nighttime) analyses. Comparing the all-sky results with those of
Figs. 3e, and f (cloud-free conditions), the $r^2$ values are similar. This is also true in terms of mean
bias, with similar values of 0.70 (-2.68) $\mu g\ m^{-3}$ found for daytime (nighttime) for all-sky scenarios.
This indicates that our method performs reasonably well from an all-sky perspective. However,
we note that restricting the analysis to solely those cases that are cloudy (not shown), the method
does not perform as well. For example, the $r^2$ values decrease by 71% (90%) and the slope values
decrease by 21% (75%) for the daytime (nighttime) analyses, compared to the cloud-free results
(Figs. 3e and f). This is expected, as any errors made in estimating the optical depths of the


overlying clouds will propagate (as biases) into the extinction retrievals for the underlying
aerosols.

**3.2.8 Aerosol type analysis**
Also, for this study, we assume that the primary aerosol type over the CONUS is pollution
(i.e., sulfate & organic) aerosol, which is generally composed of smaller (fine mode) particles that
tend to exhibit mass extinction efficiencies ~4 $m^2 g^{-1}$.  However, even after implementing our dust-
free restriction, the study region can also be contaminated with non-pollution aerosols, which can
have a larger particle size and exhibit lower mass extinction efficiencies (e.g., Hess et al., 1998;
Malm and Hand, 2007; Lynch et al., 2016).  The use of $PM_{2.5}$ versus $PM_{10}$ somewhat mitigates
this size dependency, but nevertheless coarse mode dust or sea salt can dominate $PM_{2.5}$ mass values
(e.g., Atwood et al., 2013).
Thus, in this section, the impact of aerosol types to the derived $PM_{2.5}$ concentrations was
explored by varying the mass scattering and absorption efficiencies and gamma values associated
with each aerosol type.  The three aerosol types chosen for this sensitivity study were dust, sea
salt, and smoke, based upon Lynch et al. (2016).  The mass scattering and absorption values for
dust and sea salt were interpolated to 0.532 μm from values at 0.450 μm and 0.550 μm from OPAC
(as was done for the sulfate case; Hess et al., 1998).  For smoke, these values were interpolated to
0.532 μm from values at 0.440 μm and 0.670 μm as provided by Reid et al. (2005) for smoke cases
over the US and Canada.  The gamma values were taken from Lynch et al. (2016) and the
references within.  These values, as well as the results from this sensitivity study, are shown in
Table 4.  If we assume all aerosols within the study region are smoke aerosols, no major changes
in the retrieved CALIOP $PM_{2.5}$ values are found.  However, significant uncertainties on the order



of ~200% (~800%) are found if sea salt (dust) aerosol mass scattering/absorption efficiencies and
gamma values are used instead.  Clearly, this study suggests that accurate aerosol typing is
necessary for future applications of CALIOP observations for surface $PM_{2.5}$ estimations.

**3.2.9 E-folding correlation length for $PM_{2.5}$ concentrations over the CONUS**
As a last study, we also estimated the spatial e-folding correlation length for $PM_{2.5}$
concentrations over the CONUS.  This provides us an estimation of the correlation between a
CALIOP-derived and actual $PM_{2.5}$ concentration for a given location as a function of distance
between the CALIOP observation and the given location.  To accomplish this, for 2008-2009 over
the CONUS, the distances and correlations (of $PM_{2.5}$ concentration) were computed for any two
EPA stations with over 50 days of daily data for the two-year period.  Results are shown in Fig.
10 as a scatterplot, with individual points in gray and the black curve representing the exponential
fit to the data.  A decrease in $PM_{2.5}$ correlation with distance between EPA stations is found, and
the e-folding length in correlation (e.g., correlation reduced to 1/e, or 0.37) is ~600 km (from an
AOD standpoint, this value is 40-400 km, as suggested by Anderson et al., 2003).
Also included in Fig. 10 are results from a corresponding regional analysis, with the red
and blue lines showing bin averages (10 km) for the Western and Eastern CONUS, respectively
(regions partitioned by the -97° longitude line).  The e-folding length is ~300 km (~700 km) for
the Western (Eastern) CONUS, indicating a much shorter correlation length for pollution over the
Western CONUS, possibly due to a more complex terrain such as mountains.  Overall these $PM_{2.5}$
e-folding lengths suggest that CALIOP-derived $PM_{2.5}$ concentrations could still have some
representative skill within a few hundred kilometers of a given location.



## 4    Conclusions


In this paper, we have demonstrated a new bulk-mass-modeling method for retrieving
surface particulate matter (PM) with particle sizes smaller than 2.5 μm ($PM_{2.5}$) using observations
acquired by the NASA Cloud-Aerosol Lidar with Orthogonal Polarization (CALIOP) instrument
from 2008-2009. For the purposes of demonstrating this concept, only regionally-averaged
parameters, such as mass scattering and absorption coefficients, and $PM_{2.5}$ to $PM_{10}$ (PM with
particle sizes smaller than 10 μm) conversion ratio, are used. Also, we assume the dominant type
of aerosols over the study region is pollution aerosols (supported by the occurrence frequencies of
aerosol types determined by the CALIOP algorithms), and exclude aerosol extinction range bins
classified as dust from the analysis. Even with the highly-averaged parameters, the results from
this paper are rather promising and demonstrate a potential for monitoring PM pollution using
active-based lidar observations. Specifically, the primary results of this study are as follows:
1. CALIOP-derived $PM_{2.5}$ concentrations of ~10-12.5 μg m$^{-3}$ are found over the eastern

contiguous United States (CONUS), with lower values of ~2.5-5 μg m$^{-3}$ over the central

CONUS. $PM_{2.5}$ values of ~10-20 μg m$^{-3}$ are found over the west coast of the CONUS,

primarily California. The spatial distribution of 2-year mean $PM_{2.5}$ concentrations derived

from near surface CALIOP aerosol data compares well to the spatial distribution of *in situ*

$PM_{2.5}$ measurements collected at the ground-based stations of the U.S. Environmental

Protection Agency (EPA). The use of nighttime CALIOP extinction to derive $PM_{2.5}$ results

in a higher correlation ($r^2$ = 0.48; mean bias = -3.3 μgm$^{-3}$) with EPA $PM_{2.5}$ than daytime

CALIOP extinction data ($r^2$ = 0.21; mean bias = -0.40 μgm$^{-3}$).

2. Correlations between CALIOP aerosol optical depth (AOD) and EPA $PM_{2.5}$ are much

lower ($r^2$ values of 0.04 and 0.13, for daytime and nighttime CALIOP AOD data,

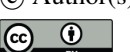


respectively) than those obtained from derived PM$_{2.5}$ using near-surface CALIOP aerosol
extinction. A similar correlation is also found between Moderate Resolution Imaging
Spectroradiometer (MODIS) AOD and EPA PM$_{2.5}$ from two-year (2008-2009) means.
This suggests that CALIOP extinction may be used as a better parameter for estimating
PM$_{2.5}$ concentrations from a 2-year mean perspective. Also, the algorithm proposed in this
study is essentially a semi-physical-based method, and thus the retrieval process can be
improved, upon a careful study of the physical parameters used in the process.
3.    Spatial and temporal sampling biases, as well as a retrieval bias, are found. Also, several
sensitivity studies were conducted, including surface layer height, cloud flag, PM$_{2.5}$/PM$_{10}$
ratio, relative humidity, and aerosol type. The sensitivity studies highlight the need for
accurate aerosol typing for estimating PM$_{2.5}$ concentrations using CALIOP observations.
4.    Using surface-based PM$_{2.5}$ at EPA stations alone, the e-folding correlation length for PM$_{2.5}$
concentrations was found to be about 600 km for the CONUS. A regional analysis yielded
values of ~300 km and ~700 km for the Western and Eastern CONUS, respectively. Thus,
while limited in spatial sampling, measurements from CALIOP may still be used for
estimating PM$_{2.5}$ concentrations over the CONUS.

As noted earlier, CALIOP observations are still rather sparse, and concerns related to

reported CALIOP aerosol extinction values also exist, such as solar and surface contamination and
the "retrieval fill value" issue (e.g., Toth et al., 2018). Yet, the future High Spectral Resolution
Lidar (HSRL) instrument on board the Earth Clouds, Aerosol, and Radiation Explorer
(EarthCARE) satellite (Illingworth et al., 2015), as well as forthcoming space-based lidar missions
in response to the 2017 Decadal Survey, offer opportunities to further explore aerosol extinction -


based PM concentrations.  Ultimately the results from this study show that the combined use of
several lidar instruments for monitoring regional and global PM pollution is potentially achievable.

























**Acknowledgements**
This research was funded with the support of the NASA Earth and Space Science Fellowship
program (NNX16A066H). Author JZ acknowledges the support from NASA grant
NNX17AG52G. CALIPSO data were obtained from the NASA Langley Research Center
Atmospheric Science Data Center (eos-web.larc.nasa.gov). MODIS data were obtained from
NASA Goddard Space Flight Center (ladsweb.nascom.nasa.gov). The PM$_{2.5}$ data were obtained
from the EPA AQS site (https://aqs.epa.gov/aqsweb/airdata/download_files.html).
















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





**Figure and Table Captions**

Figure 1. For 2008-2009, scatterplot of mean $PM_{2.5}$ concentration from ground-based U.S. EPA
stations and mean column AOD (550 nm) from collocated Collection 6 (C6) Aqua MODIS
observations.

Figure 2. For 2008-2009 over the CONUS, (a) mean $PM_{2.5}$ concentration ($\mu g\ m^{-3}$) for those U.S.
EPA stations with reported daily measurements, and (c) 1° x 1° average CALIOP-derived $PM_{2.5}$
concentrations for the 100–1000 m AGL atmospheric layer, using Equation 3, for combined
daytime and nighttime conditions. Also shown are the pairwise $PM_{2.5}$ concentrations from (b)
EPA daily measurements and (d) those derived from CALIOP (day and night combined), both
averaged for each EPA station for the 2008-2009 period. For all four plots, values greater than 20
$\mu gm^{-3}$ are colored red.

Figure 3. For 2008-2009 over the CONUS, 1° x 1° average CALIOP extinction, relative to the
number of cloud-free 5 km CALIOP profiles in each 1° x 1° bin, for the 100 – 1000 m AGL
atmospheric layer, for (a) daytime and (b) nighttime measurements. Also shown are the
corresponding CALIOP-derived $PM_{2.5}$ concentrations, using Equation 3 for (c) daytime and (d)
nighttime conditions. Values greater than 0.2 $km^{-1}$ and 20 $\mu g\ m^{-3}$ for (a, b) and (c, d), respectively,
are colored red. Scatterplots of mean $PM_{2.5}$ concentration from ground-based U.S. EPA stations
and those derived from collocated near-surface CALIOP observations are shown in the bottom
row, using (e) daytime and (f) nighttime CALIOP data.





Figure 4.  Scatterplot of mean $PM_{2.5}$ concentration from ground-based U.S. EPA stations and those
derived from collocated near-surface CALIOP observations using combined daytime and
nighttime CALIOP data.

Figure 5.  Root-mean-square errors of CALIOP-derived $PM_{2.5}$ against EPA $PM_{2.5}$ as a function of
CALIOP-derived $PM_{2.5}$ (filled circles), and corresponding number of data samples per bin (X
symbols), using both daytime (in red) and nighttime (in blue) CALIOP observations.

Figure 6.  Two-year (2008-2009) histograms of mean $PM_{2.5}$ concentrations from the U.S. EPA (in
black) and those derived from aerosol extinction using nighttime (in blue) and daytime (in red)
CALIOP data.  The U.S. EPA data shown are not collocated, while those derived using CALIOP
are spatially (but not temporally) collocated, with EPA station observations.

Figure 7.  Two-year (2008-2009) histograms of mean $PM_{2.5}$ concentrations from the U.S. EPA and
those derived from spatially and temporally collocated aerosol extinction using (a) daytime and
(b) nighttime CALIOP data.

Figure 8.  For 2008-2009, scatterplots of mean $PM_{2.5}$ concentration from ground-based U.S. EPA
stations and mean column AOD from collocated CALIOP observations, using (a) daytime and (b)
nighttime CALIOP data.




Figure 9.  For 2008-2009, scatterplots of mean $PM_{2.5}$ concentration from ground-based U.S. EPA
stations and those derived from collocated all-sky (including cloud-free and cloudy profiles) near-
surface CALIOP observations, using (a) daytime and (b) nighttime CALIOP data.

Figure 10.  For 2008-2009 over the CONUS, scatterplot of distance (km) between any two U.S.
EPA stations and the corresponding spatial correlation of $PM_{2.5}$ concentration between each pair
of stations.  The black curve represents the exponential fit to the data for the entire CONUS, and
the red and blue dashed lines represent 10 km bin averages for the Western and Eastern CONUS,
respectively.

Table 1.  The parameters, and corresponding values, used to quality assure the CALIOP aerosol
extinction profile.

Table 2.  Statistical summary of a sensitivity analysis varying the height of the surface layer,
including $R^2$, slope, mean bias (CALIOP -  EPA) of $PM_{2.5}$ in $\mu g\ m^{-3}$, and percent error change in
derived $PM_{2.5,}$ defined as: ((mean new $PM_{2.5}$ – mean original $PM_{2.5}$)/mean original $PM_{2.5}$)*100.
The row in bold represents the results shown in the remainder of the paper.

Table 3.  Statistical summary of a sensitivity analysis varying the $PM_{2.5}$ to $PM_{10}$ ratio used,
including slope, mean bias (CALIOP - EPA) of $PM_{2.5}$ in $\mu g\ m^{-3}$, and percent error change in
derived $PM_{2.5,}$ defined as: ((mean new $PM_{2.5}$ – mean original $PM_{2.5}$)/mean original $PM_{2.5}$)*100.
The row in bold represents the results shown in the remainder of the paper.

Table 4. Statistical summary of a sensitivity analysis varying the aerosol type assumed in the
derivation of PM$_{2.5}$, including R$^2$, slope, mean bias (CALIOP - EPA) of PM$_{2.5}$ in μg m$^{-3}$, and
percent error change in derived PM$_{2.5}$, defined as: ((mean new PM$_{2.5}$ – mean original PM$_{2.5}$)/mean
original PM$_{2.5}$)*100. The row in bold represents the results shown in the remainder of the paper.





















**Figures**

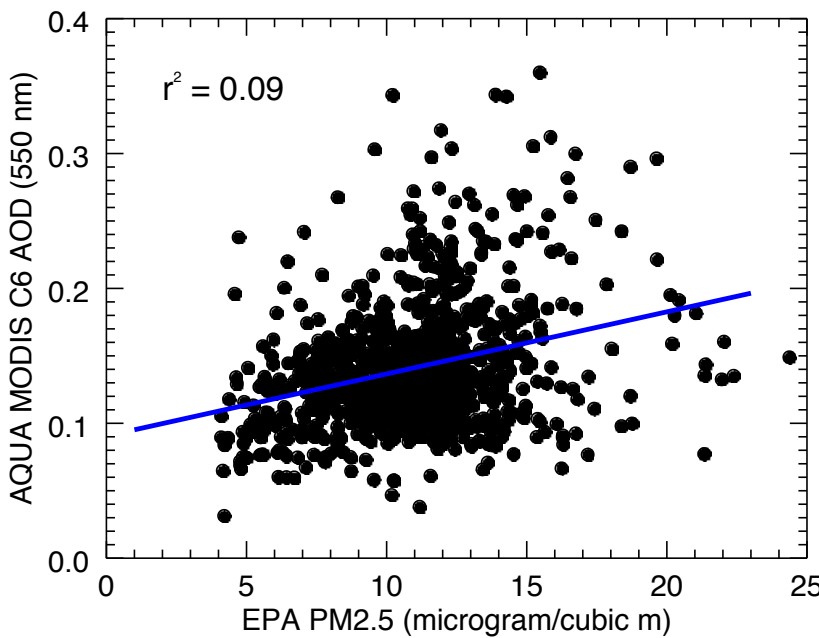

Figure 1. For 2008-2009, scatterplot of mean $PM_{2.5}$ concentration from ground-based U.S. EPA stations and mean column AOD (550 nm) from collocated Collection 6 (C6) Aqua MODIS observations.








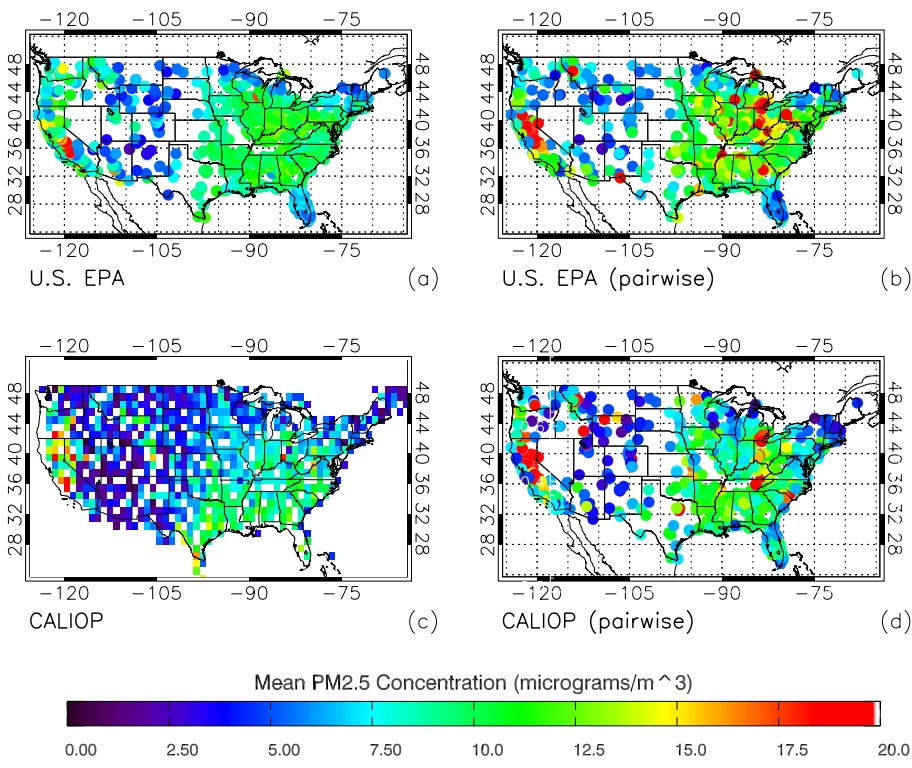

Figure 2. For 2008-2009 over the CONUS, (a) mean PM$_{2.5}$ concentration (μg m$^{-3}$) for those U.S. EPA stations with reported daily measurements, and (c) 1° x 1° average CALIOP-derived PM$_{2.5}$ concentrations for the 100–1000 m AGL atmospheric layer, using Equation 3, for combined daytime and nighttime conditions. Also shown are the pairwise PM$_{2.5}$ concentrations from (b) EPA daily measurements and (d) those derived from CALIOP (day and night combined), both averaged for each EPA station for the 2008-2009 period. For all four plots, values greater than 20 μg m$^{-3}$ are colored red.





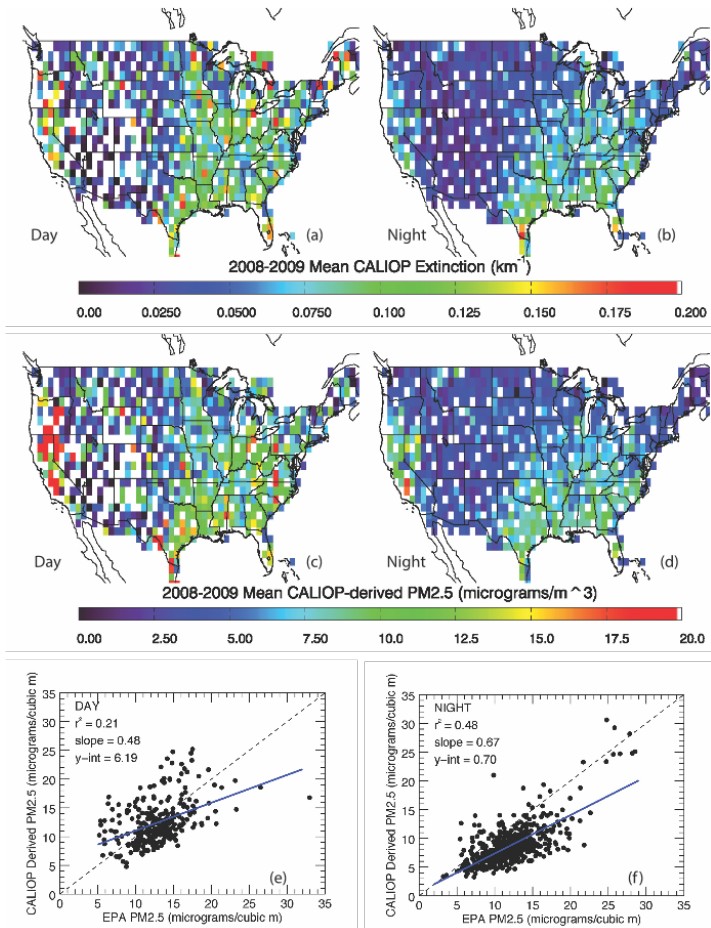

Figure 3. For 2008-2009 over the CONUS, 1° x 1° average CALIOP extinction, relative to the number of cloud-free L2_05kmAPro profiles in each 1° x 1° bin, for the 100 – 1000 m AGL atmospheric layer, for (a) daytime and (b) nighttime measurements. Also shown are the corresponding CALIOP-derived PM$_{2.5}$ concentrations, using Equation 3 for (c) daytime and (d) nighttime conditions. Values greater than 0.2 km$^{-1}$ and 20 µg m$^{-3}$ for (a, b) and (c, d), respectively, are colored red. Scatterplots of mean PM$_{2.5}$ concentration from ground-based U.S. EPA stations and those derived from collocated near-surface CALIOP observations are shown in the bottom row, using (e) daytime and (f) nighttime CALIOP data.




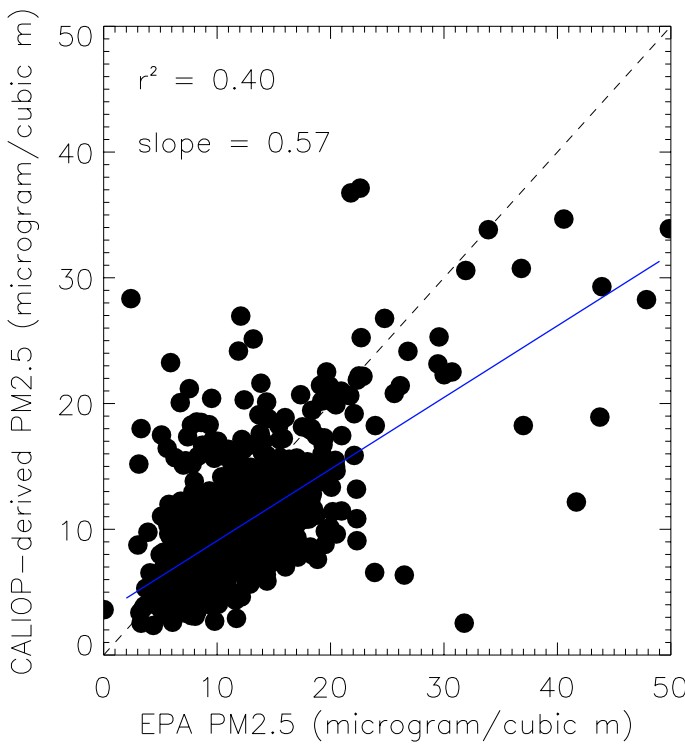

Figure 4. Scatterplot of mean PM$_{2.5}$ concentration from ground-based U.S. EPA stations and those derived from collocated near-surface CALIOP observations using combined daytime and nighttime CALIOP data.






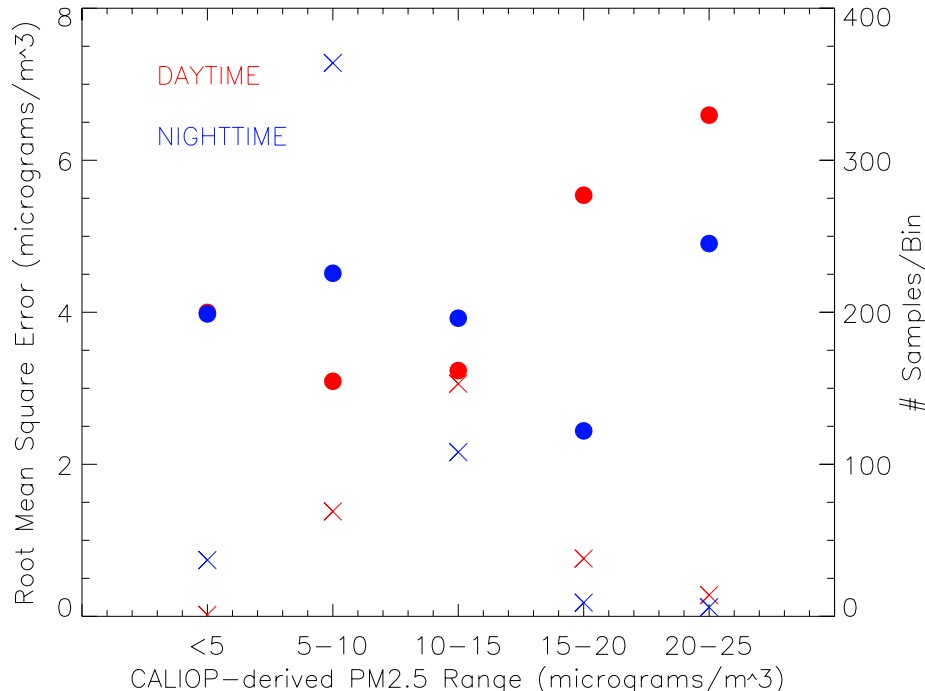

Figure 5. Root-mean-square errors of CALIOP-derived PM$_{2.5}$ against EPA PM$_{2.5}$ as a function of CALIOP-derived PM$_{2.5}$ (filled circles), and corresponding number of data samples per bin (X symbols), using both daytime (in red) and nighttime (in blue) CALIOP observations.






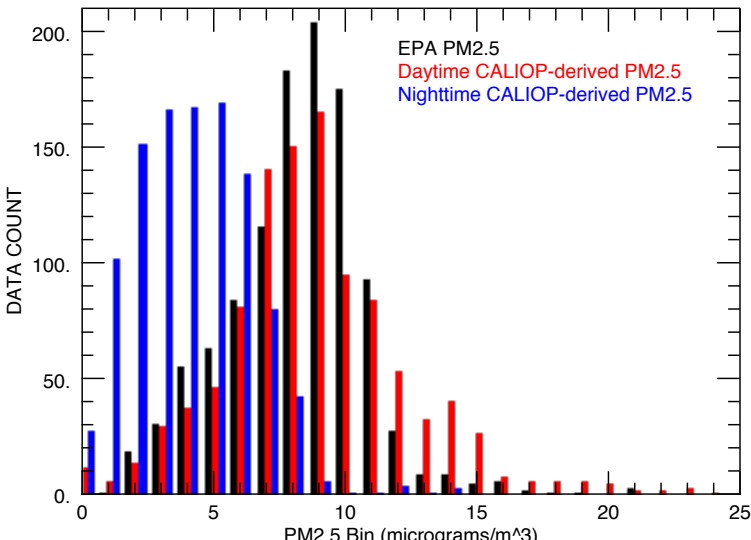

Figure 6. Two-year (2008-2009) histograms of mean PM$_{2.5}$ concentrations from the U.S. EPA (in black) and those derived from aerosol extinction using nighttime (in blue) and daytime (in red) CALIOP data. The U.S. EPA data shown are not collocated, while those derived using CALIOP are spatially (but not temporally) collocated, with EPA station observations.




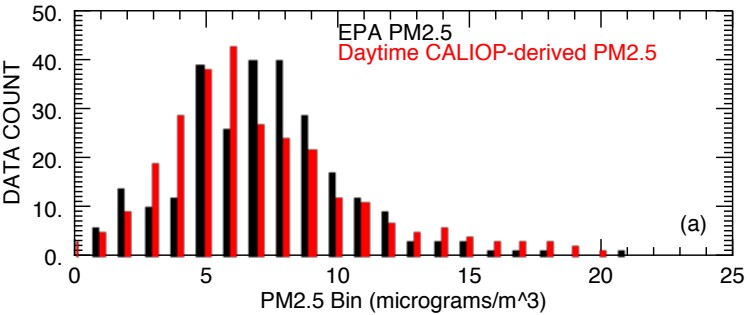

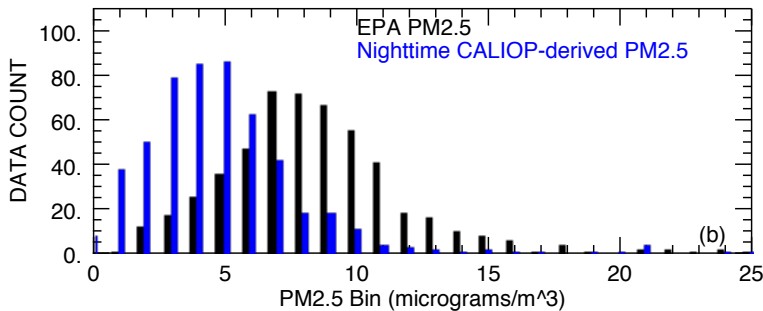

Figure 7. Two-year (2008-2009) histograms of mean PM$_{2.5}$ concentrations from the U.S. EPA and those derived from spatially and temporally collocated aerosol extinction using (a) daytime and (b) nighttime CALIOP data.




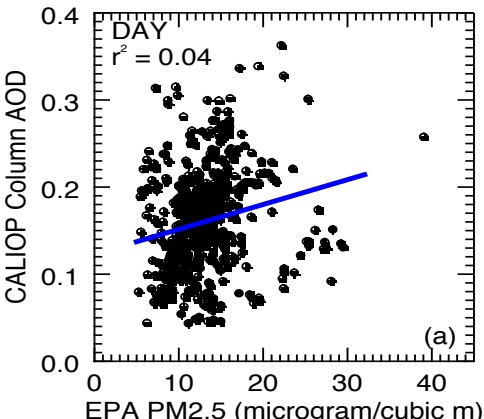

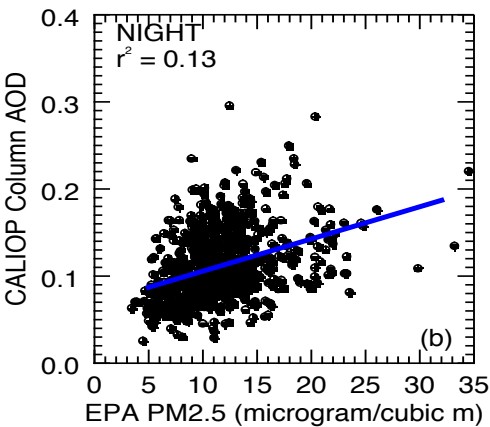

Figure 8. For 2008-2009, scatterplots of mean PM$_{2.5}$ concentration from ground-based U.S. EPA stations and mean column AOD from collocated CALIOP observations, using (a) daytime and (b) nighttime CALIOP data.






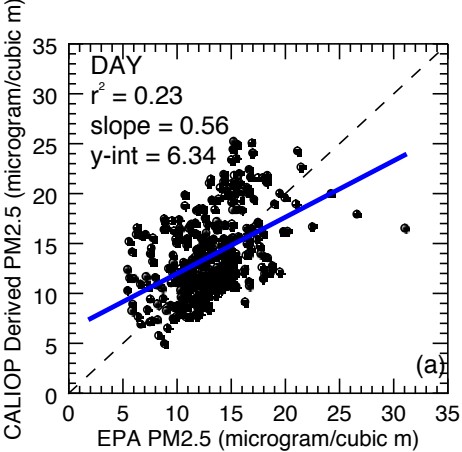

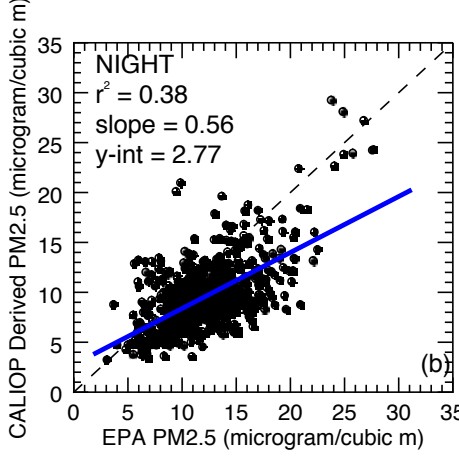

Figure 9. For 2008-2009, scatterplots of mean PM$_{2.5}$ concentration from ground-based U.S. EPA stations and those derived from collocated all-sky (including cloud-free and cloudy profiles) near-surface CALIOP observations, using (a) daytime and (b) nighttime CALIOP data.






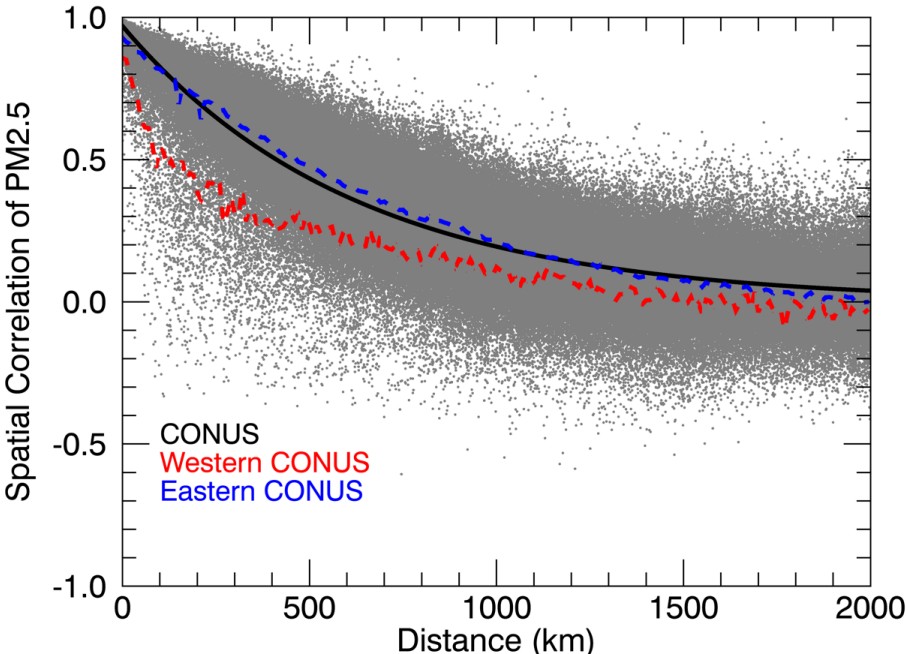

Figure 10. For 2008-2009 over the CONUS, scatterplot of distance (km) between any two U.S. EPA stations and the corresponding spatial correlation of PM$_{2.5}$ concentration between each pair of stations. The black curve represents the exponential fit to the data for the entire CONUS, and the red and blue dashed lines represent 10 km bin averages for the Western and Eastern CONUS, respectively.






**Tables**

| Parameter | Values |
|---|---|
| Integrated_Attenuated_Backscatter_532 | $\leq 0.01\ \text{sr}^{-1}$ |
| Extinction_Coefficient_532 | $\geq 0$ and $\leq 1.25\ \text{km}^{-1}$ |
| Extinction_QC_532 | $= 0, 1, 2, 16,$ or $18$ |
| CAD_Score | $\geq -100$ and $\leq -20$ |
| Extinction_Coefficient_Uncertainty_532 | $\leq 10\ \text{km}^{-1}$ |
| Atmospheric_Volume_Description (Bits 1-3) | $= 3$ |
| Atmospheric_Volume_Description (Bits 10-12) | $\neq 0$ |

Table 1. The parameters, and corresponding values, used to quality assure the CALIOP aerosol extinction profile.





| Surface Layer (m) | Analysis (Day/Night) | | | |
| --- | --- | --- | --- | --- |
| | $R^2$ | Slope | Mean Bias (CALIOP - EPA; μg m$^{-3}$) | Error Change (%) |
| 0-100 | 0.27/0.41 | 0.69/0.38 | -2.67/-9.06 | -13.71/-61.94 |
| 0-200 | 0.33/0.53 | 0.77/0.75 | -0.52/-5.68 | 3.79/-23.58 |
| 0-300 | 0.35/0.54 | 0.78/0.82 | -0.09/-4.70 | 7.24/-12.15 |
| 0-400 | 0.38/0.57 | 0.80/0.85 | -0.13/-4.25 | 6.92/-6.46 |
| 0-500 | 0.35/0.52 | 0.75/0.76 | -0.21/-4.04 | 5.70/-4.39 |
| 0-600 | 0.40/0.53 | 0.76/0.75 | -0.46/-3.91 | 3.72/-2.15 |
| 0-700 | 0.44/0.46 | 0.80/0.66 | -0.41/-3.89 | 2.73/-2.88 |
| 0-800 | 0.35/0.50 | 0.62/0.66 | -0.59/-3.76 | -0.77/-2.04 |
| 0-900 | 0.17/0.49 | 0.43/0.63 | -0.74/-3.74 | -3.91/-2.25 |
| 0-1000 | 0.13/0.48 | 0.35/0.62 | -1.08/-3.74 | -7.48/-2.57 |
| 100-500 | 0.34/0.44 | 0.72/0.66 | 0.54/-3.40 | 14.21/-0.84 |
| **100-1000** | **0.21/0.48** | **0.48/0.67** | **-0.39/-3.34** | |

Table 2. Statistical summary of a sensitivity analysis varying the height of the surface layer, including $R^2$, slope, mean bias (CALIOP - EPA) of PM$_{2.5}$ in μg m$^{-3}$, and percent error change in derived PM$_{2.5}$, defined as: ((mean new PM$_{2.5}$ – mean original PM$_{2.5}$)/mean original PM$_{2.5}$)*100. The row in bold represents the results shown in the remainder of the paper.



| | | Analysis (Day/Night) | |
| PM$_{2.5}$/PM$_{10}$ Ratio | Slope | Mean Bias (CALIOP - EPA; μg m$^{-3}$) | % Error Change |
|---|---|---|---|
| Low ratio (-1 STDEV) = 0.24 | 0.19/0.27 | -7.81/-8.61 | -60.00%/-60.00% |
| High ratio (+1 STDEV) = 0.88 | 0.71/0.98 | 5.39/0.77 | 46.67%/46.67% |
| **0.6** | **0.48/0.67** | **-0.39/-3.34** | |

Table 3. Statistical summary of a sensitivity analysis varying the PM$_{2.5}$ to PM$_{10}$ ratio used, including slope, mean bias (CALIOP - EPA) of PM$_{2.5}$ in μg m$^{-3}$, and percent error change in derived PM$_{2.5}$, defined as: ((mean new PM$_{2.5}$ – mean original PM$_{2.5}$)/mean original PM$_{2.5}$)*100. The row in bold represents the results shown in the remainder of the paper.





| Analysis (Day/Night) | | | | | | |
|---|---|---|---|---|---|---|
| **Aerosol Type** | | | **R$^2$** | **Slope** | **Mean Bias (CALIOP-EPA; μg m$^{-3}$)** | **% Error Change** |
| | $a_{scat}$ | $a_{abs}$ | $\Gamma$ | | | |
| Smoke | 5.26 | 0.26 | 0.18 | 0.10/0.44 | 0.27/0.52 | -1.81/-4.26 | -11.53/-10.54 |
| Sea salt | 1.42 | 0.01 | 0.46 | 0.18/0.48 | 1.22/1.82 | 22.42/12.93 | 184.12/184.99 |
| Dust | 0.52 | 0.08 | 0 | 0.05/0.39 | 2.06/5.12 | 102.04/70.82 | 826.94/843.33 |
| **Sulfate** | **3.4** | **0.37** | **0.63** | **0.21/0.48** | **0.48/0.67** | **-0.39/-3.34** | |

Table 4. Statistical summary of a sensitivity analysis varying the aerosol type assumed in the derivation of PM$_{2.5}$, including R$^2$, slope, mean bias (CALIOP - EPA) of PM$_{2.5}$ in μg m$^{-3}$, and percent error change in derived PM$_{2.5}$, defined as: ((mean new PM$_{2.5}$ – mean original PM$_{2.5}$)/mean original PM$_{2.5}$)*100. The row in bold represents the results shown in the remainder of the paper.
