# Peer review of "A bulk-mass-modeling-based method for retrieving Particulate Matter Pollution using"

_Atmospheric Measurement Techniques, 2018_

## Referee Comment (RC1) · Anonymous Referee #1 · 24 Oct 2018

This study represents a credible attempt at a new way to infer surface PM2.5 levels from CALIOP data, on a regional, two-year average basis. An advantage of CALIOP over passive sensors for this sort of analysis is the fact that it measures vertical profiles of backscatter and depolarisation, so bypasses a limitation inherent with imager data in partitioning between total column and near-surface aerosol loadings. In contrast, an acknowledged limitation is the curtain sampling of CALIOP vs. the broad-swath sampling of MODIS, etc. The authors introduce their technique and explain the relevant assumptions, and show results over the USA, evaluated with EPA monitors. This is a sensible, strong first step in this direction. The topic is important and relevant to AMT.

I have a number of comments (below) but on the whole recommend that the paper can be accepted after minor revisions. Hopefully this will be a springboard for further studies refining the technique and expanding to other regions and time periods.

As a general comment, much of the quantitative evaluation is presented as scatter plots with linear regression fits, and the discussion is often framed in terms of r2 and slope. I'm not sure that this is the right thing to do here. One reason is that my understanding is that there can be non-negligible uncertainties on the PM data. Indeed, Avers (2001, https://www.sciencedirect.com/science/article/pii/S1352231000005276) recommends using reduced major axis (RMA) regression instead of ordinary least squares when comparing PM monitors, for that reason. But also, the analysis in section 3 indicates that the CALIOP-derived estimates seem to have PM-dependence on their uncertainties too, so standard RMA may not be right either (as that assumes independent identically-distributed errors). For this reason I'd recommend Deming regression as a reasonable alternative (https://en.wikipedia.org/wiki/Deming regression ) when trying to compute the best-fit line. This should be more appropriate for this case, has packages in standard programming languages (and is not hard to code anyway), and is not hard to interpret. So this should be a pretty straightforward change to make which would improve the rigor of the manuscript. I recommend this is done throughout. Or, alternatively, don't fit a line but report something like mean ratio and RMS across certain ranges by binning the data.

I think it is important that appropriate statistical methods be used; continued publication using techniques we know to be deficient for our analyses just normalizes and encourages bad practice in the future. There isn't really a good justification for not fixing this.

My remaining comments are given as PXX, LYYY referring to page and line numbers respectively.

P1L21: I suggest replacing "sizes" with "diameters", as that is my understanding of

**AMTD**
the definition, but the remote sensing community often refers to radius instead when discussing size.

P4L95: I am curious as to why, with over 10 years of data, the two-year period 2008-2009 is used here? If sampling is a limiting factor in some areas, surely adding a few more years would help with this? Is there something special about these two years, or some a priori reason why two years provides sufficient sampling? I realize that running the whole mission is probably not feasible at this stage. But I would imaging that in the time between this comment being posted and the close of Open Discussion, there would be sufficient time to download and analyze an additional few years of data. This should mostly be a matter of storage and CPU time, since the code is already written (and since the first author is at Langley where CALIPSO is based, I doubt computational concerns would be significant here).

P6L123: somewhere in this initial paragraph, I'd ideally like some more discussion of the EPA data. For example, what are the uncertainties, is there any significant difference in these between the TEOM and BAM methods, and is there a difference in the siting of these two instrument types? If they're super-accurate and precise and equivalent, that's important to know. But if one is better than the other, and there's some spatial/temporal clustering in when TEOM vs. BAM is employed, that is also important to know. Recently, Kiss et al (2017, https://www.atmos-meas-tech.net/10/2477/2017/) published an analysis showing biases in hourly PM10 measurements. Is that relevant here? It might be, especially since that some daily averages in the EPA data correspond to a single sample. These are examples of things I'd like to see covered in the opening part of this section.

P8L189-190: This assumption (negligible mass above 10 micron size) is probably reasonable. But it would be fairly easy to try and quantify with AERONET. Take the inversion product from a half-dozen AERONET sites and count the fraction of the volume size distribution above 10 microns (and note here that the AERONET retrievals report size in terms of radius, while PM definitions are in diameter). You have to make some
assumption about the density of particles being the same across the size range, but otherwise that gives a first order estimate at how big the effect might be, which could be compared to the other parts of the uncertainty analysis in section 3.2. I think AERONET dust radius peaks somewhere like 2.5 microns so in the western US, it might be that there's some dust contribution from the tail of the distribution which is being systematically missed here and would lead to an overestimate in the CALIOP-derived PM levels. Maybe it is negligible, but it would be fairly easy to show that it is negligible, and the authors have not.

P13L299: An alternative to this (whether for the sensitivity analysis or the analysis as a whole) might be to look at the whole boundary layer (determining on a case by case basis) rather than testing different height ranges. Assuming that boundary layer depth is included as part of the MERRA2 meteorology being used here? This would go from assuming "the surface level of PM is represented well by the atmospheric layer from 0.1-1 km" to assuming "the boundary layer is well-mixed so represents the surface PM well", which is subtly different and might work better. I do agree that it seems reasonable to exclude the lowest 100 m, though.

P14L323: This section made me wonder why the authors do not estimate PM10 from CALIOP, and evaluate that, in addition to PM2.5? This would remove the need for an assumption of the ratio (taken as 0.6 here), and line 326 notes that there are 409 EPA stations providing both data on a daily basis. Given that this ratio seems to be one of the more uncertain parts of the error budget, it might be that there is more skill in predicting PM10 from CALIOP. Or it might go the other way. That would also be a worthwhile result, since right now we don't know.

P17L382: No particular comment here other than to say I am glad that the authors included this specific analysis. It's a point well-made that CALIOP uncertainties propagate downwards so, while CALIOP can see through thin clouds, that does not mean that the data quality is the same as for cloud-free columns.
P19L424: This isn't really an uncertainty analysis, so I suggest promoting it from a section 3.2.9 to a section 3.3 by itself. I also have a few suggestions for expansion of this section. It's good to know the correlation lengths across the western vs. eastern USA, but there's a lot of scatter in the plots. Some of this is probably due to limited sampling but some is probably also due to real changes in correlation length. So I wonder if the authors can pull out data from one or two large cities, and one or two remote areas, and highlight the correlation lengths for these (as well as the more general case of east vs. west). This would provide a bit more context about typical correlation lengths in these conditions, which would be helpful for future research built around this analysis.

**AMTD**

---

## Referee Comment (RC2) · Anonymous Referee #2 · 3 Nov 2018

**Review of the manuscript amt-2018-335**

The scope of the submitted work is to investigate the potential exploitation of CALIOP extinction profiles in order to derive near-surface concentrations of particles with aerodynamic diameter less than 2.5 μm ($PM_{2.5}$). The assessment of the applied methodology is made through the evaluation of the CALIOP derived PM concentrations against corresponding daily ground-based measurements obtained at numerous EPA stations, over the period 2008-2009, distributed across CONUS, which is the area of interest. A powerful element of using vertically resolved retrievals is that the altitude range can be constrained (i.e., near surface where the PM concentrations are measured from the ground) in contrast to passive sensors which are representative for the whole atmospheric column. To my opinion, the issues addressed by the authors fit well to the scientific objectives of AMT and therefore I recommend the submitted manuscript to be published. Nevertheless, I believe that several points must be modified making the text acceptable for publication. My major and minor comments are listed below.

**Major comments:**

1. The authors have used only 2-year satellite data thus making the robustness of the obtained outcomes questionable taking into account CALIOP's low sampling frequency and narrow footprint. In order to overcome this drawback, you have to repeat the analysis for the full dataset.
2. According to the applied methodology, all the aerosol extinctions assigned as dust in the CALIOP retrieval algorithm are masked out since focus is given on the small size particles (Lines 198-200). However, which is the treatment for the other aerosol subtypes consisting of coarse particles (i.e., marine, dusty marine)? Moreover, what is happening when the aerosol subtype is clean continental? I would suggest to repeat the aerosol type analysis (Section 3.2.8) but considering only the CALIOP aerosol subtypes which are not associated with large size particles (i.e., dust, marine, marine dust) and are relevant to pollution. Keep in mind that appropriate modifications, depending on aerosol types, may be needed in equations 1, 2 and 3 (i.e., mass scattering and absorption efficiencies, hygroscopic growth factor).
3. Could you please comment why the quality assurance criteria applied here are different than those suggested by Tacket et al. (2018; https://www.atmos-meas-tech.net/11/4129/2018/)?
4. **Page 7 – Lines 157-160:** The inclusion of different PM measurements techniques (filter-based or averages from hourly samples) how can affect the intercomparison results?
5. **Page 4 – Lines 97-102:** How much reliable are the scatterplot metrics when MODIS provides daylight AODs while PM concentrations are daily averages? Have you noticed any variation both in spatial and temporal terms?
6. **Page 9 – Line 202:** A couple of citations are needed here in order to support this argument.
7. **Page 10 – Lines 236-238:** It will be useful to provide a map with the number of days participating for the calculation of the average maps illustrated in Figure 3. Moreover, it is required a geographical distribution providing the average number of profiles considered for the derivation of 1° x 1° grid cells (i.e. an indicator of spatial representativeness within the 1deg grid cell).
8. **Page 12 – Lines 270-279:** I don't agree with the collocation criteria applied here. The horizontal distance (100 km) between CALIOP and PM station probably is too long since the analysis focuses on $PM_{2.5}$ originating from pollution. Under these cases it is expected that the horizontal variability will be very strong and the concentrations will decrease rapidly for increasing distance from the source. As it concerns the temporal collocation, the optimum solution would be to use PM measurements available at the finest temporal resolution thus making feasible an appropriate matching with the CALIOP near-surface profiles. On the contrary, if the ground-based data are provided only as daily averages then you cannot consider that a satellite overpass and a daily average are temporally collocated. In the former

data you have an instantaneous observation while in the latter one the diurnal variation is included. In case where the EPA data are given only on a daily basis, then it is more convenient to compare "daily" CALIOP profiles (considering dates where both the daytime and nighttime satellite retrievals are available) against the corresponding surface $PM_{10}$ concentrations. For this reason, I believe that Figures 3-e and 3-f as well as the relevant parts of the text must be removed. Please consider this comment throughout your analysis.

9.  **Section 3.2.1:** Considering my previous comment, the analysis should be presented only for the "daily" CALIOP – PM pairs and not separately for daytime and nighttime. Likewise, the CALIOP derived $PM_{2.5}$ ranges (x axis in Figure 5) should be equally sampled and not grouped based on user-defined bins of PM concentrations. In addition, the authors are stating in Lines 314-316 that the computations have not been done for PM concentrations $\geq 25$ µg m$^{-3}$ due to the limited number of concurrent annual means. However, according to Figure 5, the number of samples for the lowest bin ($< 5$ µg m$^{-3}$) during daytime is almost zero (the same is valid for the highest bins, particularly for the nighttime retrievals). Is that correct? Can we trust the calculated RMSEs resulting from a very small number of samples?

10. **Section 3.2.2:** To my opinion this sensitivity study should be the first step of the analysis in order to define the most "representative" altitude range. According to the summary statistics presented in Table 2, it seems that it is better to restrict the upper bound at 600 – 700m.

11. **Section 3.2.4:** Which is the impact on the $r^2$ values?

12. **Section 3.2.5:** Instead of presenting daytime and nighttime CALIOP derived PM concentrations it is better to consider only the daily (computed from the concurrent daytime and nighttime profiles) ones (see comment 6).

13. **Page 19 – Lines 448-450:** This means that the CALIOP derived PM concentrations are not reliable in coastal (contamination by sea-salt particles) or dust affected regions?

14. **Section 3.2.9:** In this section it would be also useful to provide a map with the distances where the 1/e value is found at each station.

**Minor comments:**

1.  **Page 3 – Lines 81-84:** Could you please explain better this sentence?
2.  **Page 4 – Lines 91-94:** It is not clear what the authors want to say here.
3.  **Page 10 – Line 244:** What do you mean exactly here? ("*…, as surface layer heights may change seasonally and diurnally.*")
4.  **Page 19 – Line 431:** Sulfate & organic or just sulfate?
5.  **Page 20 – Lines 456-458:** Please rephrase this sentence.

---

## Author Comment (AC1) · 25 Feb 2019

Response to Anonymous Referee #1

Comment: This study represents a credible attempt at a new way to infer surface PM2.5 levels from CALIOP data, on a regional, two-year average basis. An advantage of CALIOP over passive sensors for this sort of analysis is the fact that it measures vertical profiles of backscatter and depolarisation, so bypasses a limitation inherent with imager data in partitioning between total column and near-surface aerosol loadings. In contrast, an acknowledged limitation is the curtain sampling of CALIOP vs. the broad-swath sampling of MODIS, etc. The authors introduce their technique and explain the

relevant assumptions, and show results over the USA, evaluated with EPA monitors. This is a sensible, strong first step in this direction. The topic is important and relevant to AMT. I have a number of comments (below) but on the whole recommend that the paper can be accepted after minor revisions. Hopefully this will be a springboard for further studies refining the technique and expanding to other regions and time periods.

Response: We thank the reviewer for his/her comments and encouragement.

Comment: As a general comment, much of the quantitative evaluation is presented as scatter plots with linear regression fits, and the discussion is often framed in terms of r2 and slope. I'm not sure that this is the right thing to do here. One reason is that my understanding is that there can be non-negligible uncertainties on the PM data. Indeed, Ayers (2001, https://www.sciencedirect.com/science/article/pii/S1352231000005276 ) recommends using reduced major axis (RMA) regression instead of ordinary least squares when comparing PM monitors, for that reason. But also, the analysis in section 3 indicates that the CALIOP-derived estimates seem to have PM-dependence on their uncertainties too, so standard RMA may not be right either (as that assumes independent identically-distributed errors). For this reason I'd recommend Deming regression as a reasonable alternative (https://en.wikipedia.org/wiki/Deming_regression ) when trying to compute the best-fit line. This should be more appropriate for this case, has packages in standard programming languages (and is not hard to code anyway), and is not hard to interpret. So this should be a pretty straightforward change to make which would improve the rigor of the manuscript. I recommend this is done throughout. Or, alternatively, don't fit a line but report something like mean ratio and RMS across certain ranges by binning the data. I think it is important that appropriate statistical methods be used; continued publication using techniques we know to be deficient for our analyses just normalizes and encourages bad practice in the future. There isn't really a good justification for not fixing this.

Response: Thank you for the comments and suggestions. As recommended, Deming regression best-fit lines were added to the scatterplots of Figs. 1, 3, 4, 8, and 9, and

the slopes computed from the Deming regression analyses were added to Tables 2, 3, and 4. Corresponding changes in regard to these figures and tables were made to the narrative, and the following was added to the end of Section 2 to describe Deming regression: "Lastly, we note that most of the results are shown in the form of scatter plots with fits from Deming regression (Deming, 1943). Due to uncertainties in PM2.5 data, we show slopes computed from Deming regression analyses instead of those from simple linear regression. Deming regression in particular is appropriate here, as it accounts for errors in both the independent and dependent variables (Deming, 1943), and has been used in past PM2.5 related studies (e.g., Huang et al., 2014)."

Comment: My remaining comments are given as PXX, LYYY referring to page and line numbers respectively.

P1L21: I suggest replacing "sizes" with "diameters", as that is my understanding of the definition, but the remote sensing community often refers to radius instead when discussing size.

Response: Thank you for this suggestion. We have made the recommended changes.

Comment: P4L95: I am curious as to why, with over 10 years of data, the two-year period 2008- 2009 is used here? If sampling is a limiting factor in some areas, surely adding a few more years would help with this? Is there something special about these two years, or some a priori reason why two years provides sufficient sampling? I realize that running the whole mission is probably not feasible at this stage. But I would imaging that in the time between this comment being posted and the close of Open Discussion, there wouid be sufficient time to download and analyze an additional few years of data. This should mostly be a matter of storage and CPU time, since the code is already written (and since the first author is at Langley where CALIPSO is based, I doubt computational concerns would be significant here).

Response: The two-year period of 2008-2009 was chosen because we wanted to be consistent with the temporal domain of our previous PM2.5 study (Toth et al., 2014). An

explanation is included in Section 2. We agree that adding more years would increase sampling, but we feel this is more appropriate for a future paper, as the purpose of this manuscript is to provide an initial demonstration of the concept. An extended analysis is planned for a forthcoming paper.

Comment: P6L123: somewhere in this initial paragraph, I'd ideally like some more discussion of the EPA data. For example, what are the uncertainties, is there any significant difference in these between the TEOM and BAM methods, and is there a difference in the siting of these two instrument types? If they're super-accurate and precise and equivalent, that's important to know. But if one is better than the other, and there's some spatial/temporal clustering in when TEOM vs. BAM is employed, that is also important to know. Recently, Kiss et al (2017, https://www.atmos-meas-tech.net/10/2477/2017/ ) published an analysis showing biases in hourly PM10 measurements. Is that relevant here? It might be, especially since that some daily averages in the EPA data correspond to a single sample. These are examples of things I'd like to see covered in the opening part of this section.

Response: Thank you for the comment. As for the Kiss et al. (2017) study, PM data with a lower temporal resolution (like 24-hour, "daily" data) are less biased compared to hourly data. Still, uncertainties in hourly data are likely to impact daily data that are averaged from hourly data. To fully quantify this issue would deserve a paper of its own. Here, as suggested by the reviewer, we have edited the discussion in this section to incorporate uncertainties of the various PM2.5 measurements and spatial representativeness of the different instruments/methods. The following was added to the text:

"Note that uncertainties have been reported for hourly PM measurements (Kiss et al., 2017). Examples of some uncertainties in these PM2.5 measurements depend upon the instrument/method used: gravimetric (e.g., transport to the lab/human error and volatization of PM during the drying process; Patashnick et al., 2001), TEOM (e.g., errors due to improper inlet tube temperature; Eatough et al., 2003), and beta attenuation monitors (e.g., changes in the sample flow rate due to variations in temperature and moisture; Spagnolo, 1989). Also, it has been found that beta attenuation monitors may be more accurate than TEOM, as TEOM can underestimate PM2.5 at low temperatures (e.g., Chung et al., 2001). Still, as suggested by Kiss et al. (2017), PM data collected over a longer period of time are much less likely to be biased. Thus, we expect lower uncertainties from data collected over 24-hours, then daily data generated by averaging hourly observations. Fully quantifying the differences from the two different PM observing methods, however, is a subject for a future study."

Comment: P8L189-190: This assumption (negligible mass above 10 micron size) is probably reasonable. But it would be fairly easy to try and quantify with AERONET. Take the inversion product from a half-dozen AERONET sites and count the fraction of the volume size distribution above 10 microns (and note here that the AERONET retrievals report size in terms of radius, while PM definitions are in diameter). You have to make some assumption about the density of particles being the same across the size range, but otherwise that gives a first order estimate at how big the effect might be, which could be compared to the other parts of the uncertainty analysis in section 3.2. I think AERONET dust radius peaks somewhere like 2.5 microns so in the western US, it might be that there's some dust contribution from the tail of the distribution which is being systematically missed here and would lead to an overestimate in the CALIOP-derived PM levels. Maybe it is negligible, but it would be fairly easy to show that it is negligible, and the authors have not.

Response: It is a nice idea but we think it might be difficult to apply the proposed idea for the US for a few reasons. Firstly, reliable AERONET volume size distributions are obtained from inversions that are performed when the 440 nm AOD is larger than 0.4 (Dubovik et al., 2006). In this study, we emphasize studying 2-year means over the US, which rarely exhibit averaged 440 nm AODs larger than 0.4. Secondly, we are only concerned with near surface (100-1000 m) aerosols for this study, but AERONET would provide values for the entire column, making such a comparison difficult. We
argue that our assumption of negligible mass above 10 microns is reasonable because dust has been excluded from the analysis, and sea salt represents a small fraction of aerosols in the 100-1000 m atmospheric layer over the US for the 2008-2009 time period (i.e., < 2%). Thus, we did not implement this change as suggested.

Comment: P13L299: An alternative to this (whether for the sensitivity analysis or the analysis as a whole) might be to look at the whole boundary layer (determining on a case by case basis) rather than testing different height ranges. Assuming that boundary layer depth is included as part of the MERRA2 meteorology being used here? This would go from assuming "the surface level of PM is represented well by the atmospheric layer from 0.1-1 km" to assuming "the boundary layer is well-mixed so represents the surface PM well", which is subtly different and might work better. I do agree that it seems reasonable to exclude the lowest 100 m, though.

Response: Thank you for this suggestion. Unfortunately, boundary layer depth is not included in the MERRA-2 meteorological profiles used for this analysis. MERRA-2 relative humidity was chosen for the paper because it was already collocated with the CALIOP aerosol profiles. A boundary layer depth analysis would not be a straightforward task, and we believe a thorough study into this important topic is best left for another paper during which our method can be further refined.

Comment: P14L323: This section made me wonder why the authors do not estimate PM10 from CALIOP, and evaluate that, in addition to PM2.5? This would remove the need for an assumption of the ratio (taken as 0.6 here), and line 326 notes that there are 409 EPA stations providing both data on a daily basis. Given that this ratio seems to be one of the more uncertain parts of the error budget, it might be that there is more skill in predicting PM10 from CALIOP. Or it might go the other way. That would also be a worthwhile result, since right now we don't know.

Response: We did not estimate PM10 from CALIOP because coarse mode aerosols exhibit vastly different mass extinction efficiency values than those of fine mode

aerosols. We have included an initial look into an analysis of coarse mode aerosols, like dust and sea salt, in Table 4, the results of which suggest that large uncertainties would arise for CALIOP-derived PM values assuming coarse mode aerosols as fine mode aerosols. In order to tackle this subject, a more thorough investigation into CALIOP/ground-based aerosol typing is necessary, and we believe this topic is outside the general scope of this paper. Comment: P17L382: No particular comment here other than to say I am glad that the authors included this specific analysis. It's a point well-made that CALIOP uncertainties propagate downwards so, while CALIOP can see through thin clouds, that does not mean that the data quality is the same as for cloud-free columns.

Response: Thank you for your thoughts on this topic.

Comment: P19L424: This isn't really an uncertainty analysis, so I suggest promoting it from a section 3.2.9 to a section 3.3 by itself. I also have a few suggestions for expansion of this section. It's good to know the correlation lengths across the western vs. eastern USA, but there's a lot of scatter in the plots. Some of this is probably due to limited sampling but some is probably also due to real changes in correlation length. So I wonder if the authors can pull out data from one or two large cities, and one or two remote areas, and highlight the correlation lengths for these (as well as the more general case of east vs. west). This would provide a bit more context about typical correlation lengths in these conditions, which would be helpful for future research built around this analysis.

Response: We agree that Section 3.2.9 is not an uncertainty analysis, and it has been changed to Section 3.3. Concerning the other suggestions, each data point on the plot represents the distance of the given two locations as well as the corresponding PM correlation computed using observations from the two locations. Thus, the datasets are rather discrete and not continuous, as the correlations can only be computed with any two locations with PM observations. Thus, correlation lengths may not be derived reliably using only one or two cities. Still, we emphasize here that this section is not

the focus of the study, and can be explored in a more careful manner in a later paper.

Papers cited:

Chung, A., Chang, D. P., Kleeman, M. J., Perry, K. D., Cahill, T. A., Dutcher, D., ... & Stroud, K: Comparison of real-time instruments used to monitor airborne particulate matter, Journal of the Air & Waste Management Association, 51(1), 109-120, 2001.

Deming, W.E.: Statistical Adjustment of Data, Wiley: New York, 1943. Dubovik, O., Sinyuk, A., Lapyonok, T., Holben, B. N., Mishchenko, M., Yang, P., Eck, T. F., Volten, H., Muñoz, O., and Veihelmann, B.: Application of spheroid models to account for aerosol particle nonsphericity in remote sensing of desert dust, J. Geophys. Res.–Atmos., 111, https://doi.org/10.1029/2005JD006619, 2006. Eatough, D. J., Long, R. W., Modey, W. K., and Eatough, N. L.: Semi-volatile secondary organic aerosol in urban atmospheres: meeting a measurement challenge, Atmospheric Environment, 37(9-10), 1277-1292, 2003.

Huang, X. H., Bian, Q., Ng, W. M., Louie, P. K., and Yu, J. Z: Characterization of PM2.5 major components and source investigation in suburban Hong Kong: a one year monitoring study, Aerosol Air Qual. Res, 14(1), 237-250, 2014.

Kiss, G., Imre, K., Molnár, Á., and Gelencsér, A.: Bias caused by water adsorption in hourly PM measurements, Atmos. Meas. Tech., 10, 2477-2484, https://doi.org/10.5194/amt-10-2477-2017, 2017.

Patashnick, H., Rupprecht, G., Ambs, J. L., and Meyer, M. B.: Development of a reference standard for particulate matter mass in ambient air, Aerosol Science & Technology, 34(1), 42-45, 2001.

Spagnolo, G. S.: Automatic instrument for aerosol samples using the beta-particle attenuation, Journal of aerosol science, 20(1), 19-27, 1989.

Toth, T. D., Zhang, J., Campbell, J. R., Hyer, E. J., Reid, J. S., Shi, Y., and Westphal, D. L.: Impact of data quality and surface-to-column representativeness on the PM2.5 /

satellite AOD relationship for the contiguous United States, Atmos. Chem. Phys., 14, 6049-6062, https://doi.org/10.5194/acp-14-6049-2014, 2014.

---

## Author Comment (AC2) · 26 Feb 2019

Response to Anonymous Referee #2

Comment: The scope of the submitted work is to investigate the potential exploitation of CALIOP extinction profiles in order to derive near-surface concentrations of particles with aerodynamic diameter less than 2.5 $\mu$m (PM2.5). The assessment of the applied methodology is made through the evaluation of the CALIOP derived PM concentrations against corresponding daily ground-based measurements obtained at numerous EPA stations, over the period 2008-2009, distributed across CONUS, which is the area of interest. A powerful element of using vertically resolved retrievals is that the altitude

range can be constrained (i.e., near surface where the PM concentrations are measured from the ground) in contrast to passive sensors which are representative for the whole atmospheric column. To my opinion, the issues addressed by the authors fit well to the scientific objectives of AMT and therefore I recommend the submitted manuscript to be published. Nevertheless, I believe that several points must be modified making the text acceptable for publication. My major and minor comments are listed below.

Response: Thank you for your thoughts and encouraging comments.

Comment: The authors have used only 2-year satellite data thus making the robustness of the obtained outcomes questionable taking into account CALIOP's low sampling frequency and narrow footprint. In order to overcome this drawback, you have to repeat the analysis for the full dataset.

Response: We agree that overcoming this sampling drawback can be achieved through extending the analysis for more than two years. However, this would be computationally expensive and is a non-trivial task. We envisioned this manuscript as a proof-of-concept study, the purpose of which is to provide an initial demonstration of the feasibility of our method. Adding other years to the analysis will be one of the focuses of forthcoming CALIOP/PM2.5 papers.

Comment: According to the applied methodology, all the aerosol extinctions assigned as dust in the CALIOP retrieval algorithm are masked out since focus is given on the small size particles (Lines 198-200). However, which is the treatment for the other aerosol subtypes consisting of coarse particles (i.e., marine, dusty marine)? Moreover, what is happening when the aerosol subtype is clean continental? I would suggest to repeat the aerosol type analysis (Section 3.2.8) but considering only the CALIOP aerosol subtypes which are not associated with large size particles (i.e., dust, marine, marine dust) and are relevant to pollution. Keep in mind that appropriate modifications, depending on aerosol types, may be needed in equations 1, 2 and 3 (i.e., mass scattering and absorption efficiencies, hygroscopic growth factor).

Response: All aerosol subtypes not classified as dust are considered for our method (e.g., marine, dusty marine, clean continental, etc.). We have already excluded dust, and most areas of the CONUS are not dominated by sea salt aerosols. Indeed, a statistical analysis showed that CALIOP 100-1000 m aerosol layers consisting entirely of marine (dusty marine) subtypes represent only ∼2% (∼1%) of all subtypes. Thus, the impact of including these aerosols should be minor. We do note, however, that one of the areas of focus for future studies of CALIOP-derived estimates is a more thorough investigation into aerosol typing.

Comment: Could you please comment why the quality assurance criteria applied here are different than those suggested by Tacket et al. (2018; https://www.atmos-meas-tech.net/11/4129/2018/)?

Response: The QA criteria applied for this paper are the same as those of our previous CALIOP papers (e.g., Toth et al. 2014; 2016; 2018), and we wanted to be consistent with these studies. The QA scheme employed here was developed from Kittaka et al. (2011) and Campbell et al. (2012), both of which provide detailed justifications for the QA choices made. Toth et al. (2016) provides comparisons of aerosol extinction profiles using our QA scheme and those from the CALIPSO Level 3 aerosol profile product. While some differences were found, these were mostly attributed to differences in averaging, treatment of clouds and fill values, and our vertical regridding from 60 m to 100 m.

Comment: Page 7 – Lines 157-160: The inclusion of different PM measurements techniques (filter-based or averages from hourly samples) how can affect the intercomparison results?

Response: As suggested by Kiss et al. (2017), large uncertainties exist in hourly PM data, while less biases are expected for PM data collected over a longer period of time. Thus, there are likely differences in the two methods for collecting PM data. Still, to fully explore this issue requires a study of its own, and thus we have added the following

discussions in the text:

"Note that uncertainties have been reported for hourly PM measurements (Kiss et al., 2017). Examples of some uncertainties in these PM2.5 measurements depend upon the instrument/method used: gravimetric (e.g., transport to the lab/human error and volatization of PM during the drying process; Patashnick et al., 2001), TEOM (e.g., errors due to improper inlet tube temperature; Eatough et al., 2003), and beta attenuation monitors (e.g., changes in the sample flow rate due to variations in temperature and moisture; Spagnolo, 1989). Also, it has been found that beta attenuation monitors may be more accurate than TEOM, as TEOM can underestimate PM2.5 at low temperatures (e.g., Chung et al., 2001). Still, as suggested by Kiss et al. (2017), PM data collected over a longer period of time are much less likely to be biased. Thus, we expect lower uncertainties from data collected over 24-hours, then daily data generated by averaging hourly observations. Fully quantifying the differences from the two different PM observing methods, however, is the subject for a future study."

Comment: Page 4 – Lines 97-102: How much reliable are the scatterplot metrics when MODIS provides daylight AODs while PM concentrations are daily averages? Have you noticed any variation both in spatial and temporal terms?

Response: In this paper, we have not looked into the spatial/temporal variations of MODIS AOD versus PM2.5. This, however, was the subject of one of our past studies (Toth et al., 2014), for which MODIS AOD was compared to both daily (within 1x1 deg.) and hourly (within 40 km) PM2.5 measurements. While larger correlation coefficients were found for the hourly analysis, they still remained low. The purpose of Fig. 1 in this paper was to simply illustrate the limitation of using column-integrated AOD from passive sensors to estimate PM2.5 concentrations near the surface.

Comment: Page 9 – Line 202: A couple of citations are needed here in order to support this argument.

Response: We have added two citations (Nessler et al., 2005 and Lynch et al., 2016),

as requested.

Comment: Page 10 – Lines 236-238: It will be useful to provide a map with the number of days participating for the calculation of the average maps illustrated in Figure 3. Moreover, it is required a geographical distribution providing the average number of profiles considered for the derivation of 1° x 1° grid cells (i.e. an indicator of spatial representativeness within the 1deg grid cell).

Response: Thank you for this suggestion. We have added the requested maps as a figure in an appendix. Also, the following description was added to the text in Section 3.1: "Note that, for context, maps of the number of days and CALIOP Level 2 5 km aerosol profiles used in the creation of Fig. 3a-d are shown in Appendix Fig. 1."

Comment: Page 12 – Lines 270-279: I don't agree with the collocation criteria applied here. The horizontal distance (100 km) between CALIOP and PM station probably is too long since the analysis focuses on PM2.5 originating from pollution. Under these cases it is expected that the horizontal variability will be very strong and the concentrations will decrease rapidly for increasing distance from the source. As it concerns the temporal collocation, the optimum solution would be to use PM measurements available at the finest temporal resolution thus making feasible an appropriate matching with the CALIOP near-surface profiles. On the contrary, if the ground-based data are provided only as daily averages then you cannot consider that a satellite overpass and a daily average are temporally collocated. In the former data you have an instantaneous observation while in the latter one the diurnal variation is included. In case where the EPA data are given only on a daily basis, then it is more convenient to compare "daily" CALIOP profiles (considering dates where both the daytime and nighttime satellite retrievals are available) against the corresponding surface PM10 concentrations. For this reason, I believe that Figures 3-e and 3-f as well as the relevant parts of the text must be removed. Please consider this comment throughout your analysis.

Response: Thanks for the suggestion. We agree that "daily" averaged CALIOP profiles

may be used for comparing with daily averaged surface PM observations. However, with a narrow swath of ∼70 m and a repeat cycle of 16 days, very few data points would be available within 100 km of a particular EPA site for both daytime and nighttime CALIOP aerosol profiles. For the spatial collocation, the +/- 100 km collocation distance is used here, as we considered the spread of aerosols within 24 hours. For example, for a 10 km/hour wind speed, aerosol particles may travel 200 km (or +/- 100 km) within 24 hours. Also, as suggested from this paper, the averaged e-folding correlation length for PM2.5 concentrations over the CONUS is ∼600 km, and thus we believe 100 km is a reasonable collocation range.

Also, analysis using finer temporal resolution PM2.5 data may produce better results under some conditions, but comes with its own issues. For example, there are insufficient collocated CALIOP profiles and hourly PM2.5 data over a two-year period for the CONUS, so the temporal domain would need to be greatly expanded. Secondly, this type of study would take careful analysis of the CALIOP data, as individual CALIOP aerosol extinction profiles could be subject to higher uncertainties (e.g., rather than using a two-year mean). These research topics will be examined in detail in future studies.

Comment: Section 3.2.1: Considering my previous comment, the analysis should be presented only for the "daily" CALIOP – PM pairs and not separately for daytime and nighttime. Likewise, the CALIOP derived PM2.5 ranges (x axis in Figure 5) should be equally sampled and not grouped based on user-defined bins of PM concentrations. In addition, the authors are stating in Lines 314-316 that the computations have not been done for PM concentrations $\geq$ 25 $\mu$g m-3 due to the limited number of concurrent annual means. However, according to Figure 5, the number of samples for the lowest bin (< 5 $\mu$g m-3) during daytime is almost zero (the same is valid for the highest bins, particularly for the nighttime retrievals). Is that correct? Can we trust the calculated RMSEs resulting from a very small number of samples?

Response: As mentioned in another response, using only "daily" CALIOP-PM pairs

is not feasible here for a robust analysis, due to the repeat cycle of the CALIPSO satellite. Very few data points would be available within 100 km of a particular EPA site for both daytime and nighttime CALIOP aerosol profiles. Thus, we leave daytime and nighttime separated for this figure. As for the CALIOP derived PM2.5 ranges, we have adjusted them such that each bin is equally sampled based upon a cumulative histogram analysis. Each point from left to right in the new Fig. 5 represents the RMSE and mean PM2.5 concentration derived from CALIOP for 0-20%, 20-40%, 40-60%, 60-80%, and 80-100% cumulative frequencies. This addresses the other items in this comment, like those of few samples for the lowest and highest bins in the old Fig. 5. Because they are now equally sampled, we have removed the secondary y-axis since the number of samples do not change as a function of CALIOP-derived PM2.5 concentration.

We have also revised the corresponding text in Section 3.2.1 as follows:

"As a first step for the uncertainty analysis, we estimated the prognostic error of 2-year averaged PM2.5_CALIOP. Figure 5 shows the root-mean-square error (RMSE) of CALIOP-based PM2.5 concentrations against those from EPA stations as a function of CALIOP-based PM2.5 for the 2008-2009 period over the CONUS. RMSEs were computed for five equally sampled bins, determined from a cumulative histogram analysis. Each point in Fig. 5, from left to right, represents the RMSE and mean PM2.5 concentration derived from CALIOP for 0-20%, 20-40%, 40-60%, 60-80%, and 80-100% cumulative frequencies. A mean combined daytime and nighttime RMSE of $\sim$4 $\mu$g m-3 is found, with a mean value slightly greater for nighttime ($\sim$4.3 $\mu$g m-3) than daytime ($\sim$3.7 $\mu$g m-3). While most bins exhibit larger nighttime RMSEs, daytime RMSEs are larger for the greatest mean CALIOP-derived PM2.5 concentrations."

Comment: Section 3.2.2: To my opinion this sensitivity study should be the first step of the analysis in order to define the most "representative" altitude range. According to the summary statistics presented in Table 2, it seems that it is better to restrict the upper bound at 600 – 700m.

Response: While a surface layer up to about 600-700 m results in larger r2 values, much variability in the statistics exists between surface layer heights (as shown in Table 2). Also, differences are found between daytime and nighttime for various layers. One possible issue is a lower signal-to-noise ratio if we restricted the surface layer to lower heights. We stress that the purpose of this paper is an initial exploration of the topic, and wanted to include Table 2 as a first look at surface layer height sensitivity. Another study is necessary to better evaluate this subject, especially as surface layer height changes regionally and diurnally.

Comment: Section 3.2.4: Which is the impact on the r2 values?

Response: The r2 values are not impacted by varying the PM2.5/PM10 ratio. This is because all of the CALIOP-derived PM2.5 points for each scenario shown in Table 3 are multiplied by a common ratio (see Equation 3), but the collocated EPA concentrations remain unchanged (thus not altering the correlation).

Comment: Section 3.2.5: Instead of presenting daytime and nighttime CALIOP derived PM concentrations it is better to consider only the daily (computed from the concurrent daytime and nighttime profiles) ones (see comment 6).

Response: Thank you for the suggestion. We believe it is important to show the daytime and nighttime analyses separately, and an analysis using concurrent daytime and nighttime profiles collocated with a particular EPA site will not yield many samples due to the repeat cycle of the CALIPSO satellite. Thus we didn't make the change.

Comment: Page 19 – Lines 448-450: This means that the CALIOP derived PM concentrations are not reliable in coastal (contamination by sea-salt particles) or dust affected regions?

Response: The large uncertainties are because mass extinction efficiencies are drastically different for coarse and fine mode aerosols. Here we applied mass extinction efficiencies from fine mode aerosols to coarse mode aerosols, and not surprisingly,

see large uncertainties. Lower uncertainties can be expected if we apply coarse mode mass extinction efficiencies to coarse mode aerosols. However, this puts the pressure on accurate estimations of aerosol types from CALIOP or other lidar observations, which we believe is a study of its own, and will be investigated in future studies.

Comment: Section 3.2.9: In this section it would be also useful to provide a map with the distances where the 1/e value is found at each station.

Response: Thank you for this suggestion. However, for each pair of PM observing locations, one correlation value is computed for a given distance between the two locations. Thus, the analysis is discrete, not continuous. The 1/e values are estimated from Fig. 10, which is composed of individual points representing both a distance and spatial PM2.5 correlation between pairs of EPA sites over the CONUS. If we apply the same analysis to a given PM observing location, it is likely to have data gaps due to the discrete nature of the dataset. Thus, we leave Fig. 10 untouched.

Comment: Page 3 – Lines 81-84: Could you please explain better this sentence?

Response: We have rewritten the sentence to:

"Indeed, Kaku et al. (2018) recently showed that surface PM2.5 had longer spatial correlation lengths than AOD, even in the "well behaved" southeastern United States where previous studies showed good correlation between PM2.5 and AOD (e.g., Wang and Christopher, 2003)."

Comment: Page 4 – Lines 91-94: It is not clear what the authors want to say here.

Response: We have rewritten the sentence to:

"It is arguable that from a climatological/long-term average perspective, the use of AOD as a proxy for PM2.5 concentrations nevertheless has certain qualitative skill (e.g., Toth et al., 2014; Reid et al., 2017) due to the averaging process that suppresses sporadic aerosol events with highly variable vertical distributions."

Comment: Page 10 – Line 244: What do you mean exactly here? ("..., as surface layer heights may change seasonally and diurnally.")

Response: We have removed "as surface layer heights may change seasonally and diurnally" to avoid confusion.

Comment: Page 19 – Line 431: Sulfate & organic or just sulfate?

Response: To avoid confusion, we removed "& organic". But primary and secondary biogenic aerosols are mostly fine mode as well.

Comment: Page 20 – Lines 456-458: Please rephrase this sentence.

Response: This sentence was broken into two sentences, as follows: "To accomplish this, all EPA stations over the CONUS with at least 50 days of daily data available for the 2008-2009 period were first determined. Next, the distances between each pair of these EPA stations, and their corresponding correlation of daily PM2.5 concentrations, were computed."

Papers cited:

Campbell, J. R., Tackett, J. L., Reid, J. S., Zhang, J., Curtis, C. A., Hyer, E. J., Sessions, W. R., Westphal, D. L., Prospero, J. M., Welton, E. J., Omar, A. H., Vaughan, M. A., and Winker, D. M.: Evaluating nighttime CALIOP 0.532 $\mu$m aerosol optical depth and extinction coefficient retrievals, Atmos. Meas. Tech., 5, 2143-2160, https://doi.org/10.5194/amt-5-2143-2012, 2012.

Chung, A., Chang, D. P., Kleeman, M. J., Perry, K. D., Cahill, T. A., Dutcher, D., ... & Stroud, K: Comparison of real-time instruments used to monitor airborne particulate matter, Journal of the Air & Waste Management Association, 51(1), 109-120, 2001.

Deming, W.E.: Statistical Adjustment of Data, Wiley: New York, 1943. Eatough, D. J., Long, R. W., Modey, W. K., and Eatough, N. L.: Semi-volatile secondary organic aerosol in urban atmospheres: meeting a measurement challenge, Atmospheric Environment, 37(9-10), 1277-1292, 2003.

Huang, X. H., Bian, Q., Ng, W. M., Louie, P. K., and Yu, J. Z: Characterization of PM2.5 major components and source investigation in suburban Hong Kong: a one year monitoring study, Aerosol Air Qual. Res, 14(1), 237-250, 2014.

Kiss, G., Imre, K., Molnár, Á., and Gelencsér, A.: Bias caused by water adsorption in hourly PM measurements, Atmos. Meas. Tech., 10, 2477-2484, https://doi.org/10.5194/amt-10-2477-2017, 2017.

Kittaka, C., Winker, D. M., Vaughan, M. A., Omar, A., & Remer, L. A.: Intercomparison of column aerosol optical depths from CALIPSO and MODIS-Aqua, Atmospheric Measurement Techniques, 4(2), 131, https://doi.org/10.5194/amt-4-131-2011, 2011.

Lynch P., and coauthors: An 11-year global gridded aerosol optical thickness reanalysis (v1.0) for atmospheric and climate sciences, Geosci. Model Dev., 9, 1489-1522, doi:10.5194/gmd-9-1489-2016, 2016.

Nessler, R., Weingartner, E., and Baltensperger, U. (2005). Effect of humidity on aerosol light absorption and its implications for extinction and the single scattering albedo illustrated for a site in the lower free troposphere, Journal of Aerosol Science, 36(8), 958-972.

Patashnick, H., Rupprecht, G., Ambs, J. L., and Meyer, M. B.: Development of a reference standard for particulate matter mass in ambient air, Aerosol Science & Technology, 34(1), 42-45, 2001.

Spagnolo, G. S.: Automatic instrument for aerosol samples using the beta-particle attenuation, Journal of aerosol science, 20(1), 19-27, 1989.

Toth, T. D., Campbell, J. R., Reid, J. S., Tackett, J. L., Vaughan, M. A., Zhang, J., and Marquis, J. W.: Minimum aerosol layer detection sensitivities and their subsequent impacts on aerosol optical thickness retrievals in CALIPSO level 2 data products, Atmos. Meas. Tech., 11, 499-514, https://doi.org/10.5194/amt-11-499-2018, 2018.

Toth, T. D., Zhang, J., Campbell, J. R., Hyer, E. J., Reid, J. S., Shi, Y., and Westphal, D. L.: Impact of data quality and surface-to-column representativeness on the PM2.5 / satellite AOD relationship for the contiguous United States, Atmos. Chem. Phys., 14, 6049-6062, https://doi.org/10.5194/acp-14-6049-2014, 2014.

Toth, T. D., Zhang, J., Campbell, J. R., Reid, J. S., and Vaughan, M. A.: Temporal variability of aerosol optical thickness vertical distribution observed from CALIOP, Journal of Geophysical Research: Atmospheres, 121(15), 9117-9139, https://doi.org/10.1002/2015JD024668, 2016.

———————————————————